# Shape-invariant encoding of dynamic primate facial expressions in human perception

**Nick Taubert[1‡], Michael Stettler[1,2‡], Ramona Siebert[3], Silvia Spadacenta[3], Louisa Sting[1], Peter Dicke[3], Peter Thier[3†], Martin A Giese[1†*]**

[1]Section for Computational Sensomotorics, Centre for Integrative Neuroscience & Hertie Institute for Clinical Brain Research, University Clinic Tübingen, Tübingen, Germany; [2]International Max Planck Research School for Intelligent Systems (IMPRS-IS), Tübingen, Germany; [3]Department of Cognitive Neurology, Hertie Institute for Clinical Brain Research, University of Tübingen, Tübingen, Germany

**Abstract** Dynamic facial expressions are crucial for communication in primates. Due to the difficulty to control shape and dynamics of facial expressions across species, it is unknown how species-specific facial expressions are perceptually encoded and interact with the representation of facial shape. While popular neural network models predict a joint encoding of facial shape and dynamics, the neuromuscular control of faces evolved more slowly than facial shape, suggesting a separate encoding. To investigate these alternative hypotheses, we developed photo-realistic human and monkey heads that were animated with motion capture data from monkeys and humans. Exact control of expression dynamics was accomplished by a Bayesian machine-learning technique. Consistent with our hypothesis, we found that human observers learned cross-species expressions very quickly, where face dynamics was represented largely independently of facial shape. This result supports the co-evolution of the visual processing and motor control of facial expressions, while it challenges appearance-based neural network theories of dynamic expression recognition.

**\*For correspondence:**
martin.giese@uni-tuebingen.de

[†]These authors contributed equally to this work
[‡]These authors also contributed equally to this work

**Competing interests:** The authors declare that no competing interests exist.

## Introduction

Facial expressions are crucial for social communication of human as well as non-human primates (*Calder, 2011*; *Darwin, 1872*; *Jack and Schyns, 2017*; *Curio et al., 2010*), and humans can learn facial expressions even of other species (*Nagasawa et al., 2015*). While facial expressions in every-day life are dynamic, specifically, expression recognition across different species has been studied mainly using static pictures of faces (*Campbell et al., 1997*; *Dahl et al., 2013*; *Sigala et al., 2011*; *Guo et al., 2019*; *Dahl et al., 2009*). A few studies have compared the perception of human and monkey expressions using movie stimuli, finding overlaps in the brain activation patterns induced by within- and cross-species expression observation in humans as well as monkeys (*Zhu et al., 2013*; *Polosecki et al., 2013*). Since natural video stimuli provide no accurate control of the dynamics and form features of facial expressions, it is unknown how expression dynamics is perceptually encoded across different primate species and how it interacts with the representation of facial shape.

In primate phylogenesis, the visual processing of dynamic facial expressions has co-evolved with the neuromuscular control of faces (*Schmidt and Cohn, 2001*). Remarkably, the structure and arrangement of facial muscles is highly similar across different primate species (*Vick et al., 2007*; *Parr et al., 2010*), while face shapes differ considerably, for example, between humans, apes, and monkeys. This motivates the following two hypotheses: (1) The phylogenetic continuity in motor control should facilitate fast learning of dynamic expressions across primate species and (2) the different

speeds of the phylogenetic development of the facial shape and its motor control should potentially imply a separate visual encoding of expression dynamics and basic face shape. The second hypothesis seems consistent with a variety of data in functional imaging, which suggests a partial separation of the anatomical structures processing changeable and non-changeable aspects of faces (*Haxby et al., 2000*; *Bernstein and Yovel, 2015*).

We investigated these hypotheses, exploiting advanced methods from computer animation and machine learning, combined with motion capture in monkeys and humans. We designed highly realistic three-dimensional (3D) human and monkey avatar heads by combining structural information derived from 3D scans, multi-layer texture models for the reflectance properties of the skin, and hair animation. Expression dynamics was derived from motion capture recordings on monkeys and humans, exploiting a hierarchical generative Bayesian model to generate a continuous motion style space. This space includes continuous interpolations between two expression types ('anger' vs. 'fear'), and human- and monkey-specific motions. Human observers categorized these dynamic expressions, presented on the human or the monkey head model, in terms of the perceived expression type and species-specificity of the motion (human vs. monkey expression).

Consistent with our hypotheses, we found very fast cross-species learning of expression dynamics with a typically narrower tuning for human- compared to monkey-specific expressions. Most importantly, the perceptual categorization of expression dynamics was largely independent of the facial shape (human vs. monkey). In particular, the accuracy of the categorization of species-specific dynamic facial expressions did not show a dependence on whether the species-specific expressive motion and the avatar species were matching (e.g., monkey expressions being recognized more accurately on a monkey avatar). Our results were highly robust against substantial variations in the expressive stimulus features. They specify fundamental constraints for the computational neural mechanisms of dynamic face processing and challenge popular neural network models, accounting for expression recognition by the learning of sequences of key shapes (e.g. *Curio et al., 2010*).

## Results

In this section, we briefly sketch the methodology of our experiments; whereas many other important details can be found in 'Materials and methods' section and 'Appendix 1'. Then, we describe in detail the results of the three main experiments, which we realized (further control experiments are described in 'Appendix 1').

Our studies investigated the perceptual representations of dynamic human and monkey facial expressions in human observers, exploiting photo-realistic human and monkey face avatars (*Figure 1A*). The motion of the avatars was generated exploiting motion capture data of both primate species (*Figure 1B*), which were used to compute the corresponding deformation of the surface mesh of the face, exploiting a model based on elastic ribbon structures that were modeled after the main facial muscles of humans and monkeys (*Figure 1C* and Appendix 1).

In order to realize a full parametric control of motion style, we exploited a Bayesian motion morphing technique ('Materials and methods') to create a continuous expression space that smoothly interpolates between human and monkey expressions. We used two human expressions and two monkey expressions as basic patterns, which represented corresponding emotional states ('fear' and 'anger/threat'). Interpolating between these four prototypical motions in five equidistant steps, we generated a set of 25 facial movements that vary in five steps along two dimensions, the expression type, and the species, as illustrated in *Figure 1D*. Each generated motion pattern can be parameterized by a two-dimensional style vector $(e, s)$, where the first component $e$ specifies the expression type ($e = 0$: expression 1 ('fear') and $e = 1$: expression 2 ('anger/threat')), and where the second variable $s$ defines the species-specificity of the motion ($s = 0$: monkey and $s = 1$: human). The dynamic expressions were used to animate a highly realistic monkey as well as a human avatar model (generation; 'Materials and methods'). In order to vary the two-dimensional stimulus features, we rendered the avatars from two different view angles: from the front view and from the view that was rotated by 30 degrees about the vertical axis. This rotated view maximized the differences of the two-dimensional appearance relative to the front view, while avoiding strong salient changes, such as occlusions of face parts. The following sections describe the results of the three main experiments of our study.

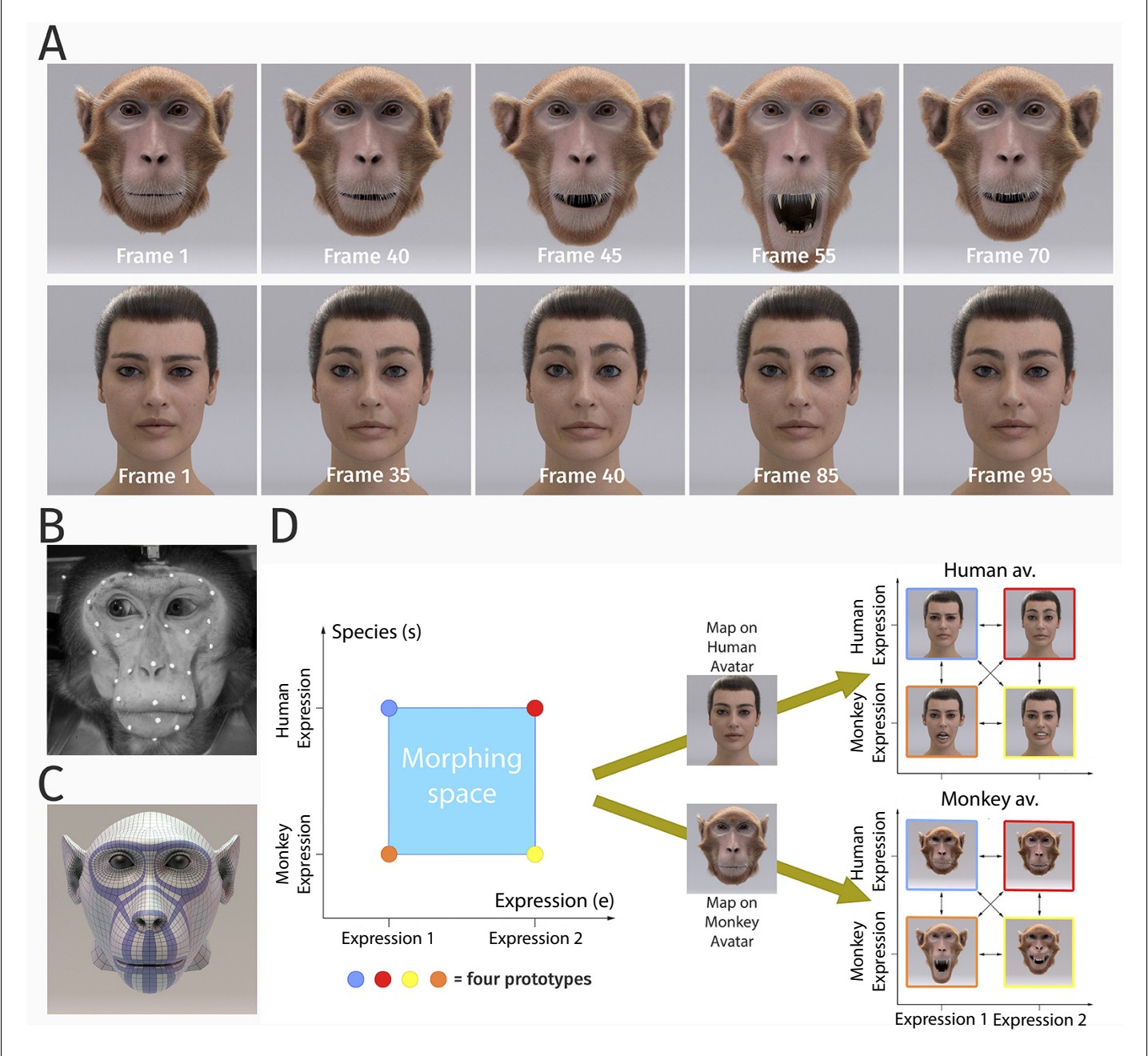

**Figure 1.** Stimulus generation and paradigm. (A) Frame sequence of a monkey and a human facial expression. (B) Monkey motion capture with 43 reflecting facial markers. (C) Regularized face mesh, whose deformation is controlled by an embedded elastic ribbon-like control structure that is optimized for animation. (D) Stimulus consisting of 25 motion patterns, spanning up a two-dimensional style space with the dimensions 'expression' and 'species', generated by interpolation between two expressions ('anger/threat' and 'fear') and the two species ('monkey' and 'human'). Each motion pattern was used to animate a monkey and a human avatar model.

## Dynamic expression perception is largely independent of facial shape

In our first experiment, we used the original dynamic expressions of humans and monkeys as prototypes and presented morphs between them, separately, on the human and the monkey avatar faces, with two different view angles (0 and 30 degrees rotation about the vertical axis). Facial movements of humans and monkeys are quite different (*Vick et al., 2007*), so that our participants, all of whom had no prior experience with macaque monkeys, needed to be familiarized briefly with the monkey

expressions prior to the main experiment. During the familiarization, participants learned to recognize the four prototypical expressions perfectly, always with maximally four stimulus repetitions. During the main experiment, motions were presented in a block-randomized order, and in separate blocks for the two avatars and for the two tested views. The expression movies with a duration of 5 s showed the face going from a neutral expression to the extreme expression and back to neutral (*Figure 1A*). Participants observed 10 repetitions of each stimulus. They had to decide whether the observed stimulus was looking more like a human or a monkey expression (independent of the avatar shape and view), and whether the expression was rather 'anger/threat' or 'fear'. The resulting two binary responses in each trial can be interpreted as assignment of one out of four classes to the perceived expression of the stimulus, independent of avatar type and view (1: human-angry, 2: human-fear, 3: monkey-threat, and 4: monkey-fear).

*Figure 2A* shows the raw classification data as histograms of the relative frequencies of the four classes $\hat{C}_i(e, s)$, as a function of the style parameters $e$ and $s$ for the four tested classes. The class probabilities $P_i(e, s)$ were modeled by a logistic multinomial regression model ('Materials and methods'), resulting in the fitted discriminant functions that are shown in *Figure 2B* for the different classes. Comparing regression models with different sets of predictor variables, we found that in almost all cases, a model of the form that contains the two style variables for expression ($e$) and the species ($s$) as predictors (in addition to a constant predictor) was the simplest model that provided good fits of the data. *Figure 2C* shows the prediction accuracy of regression models

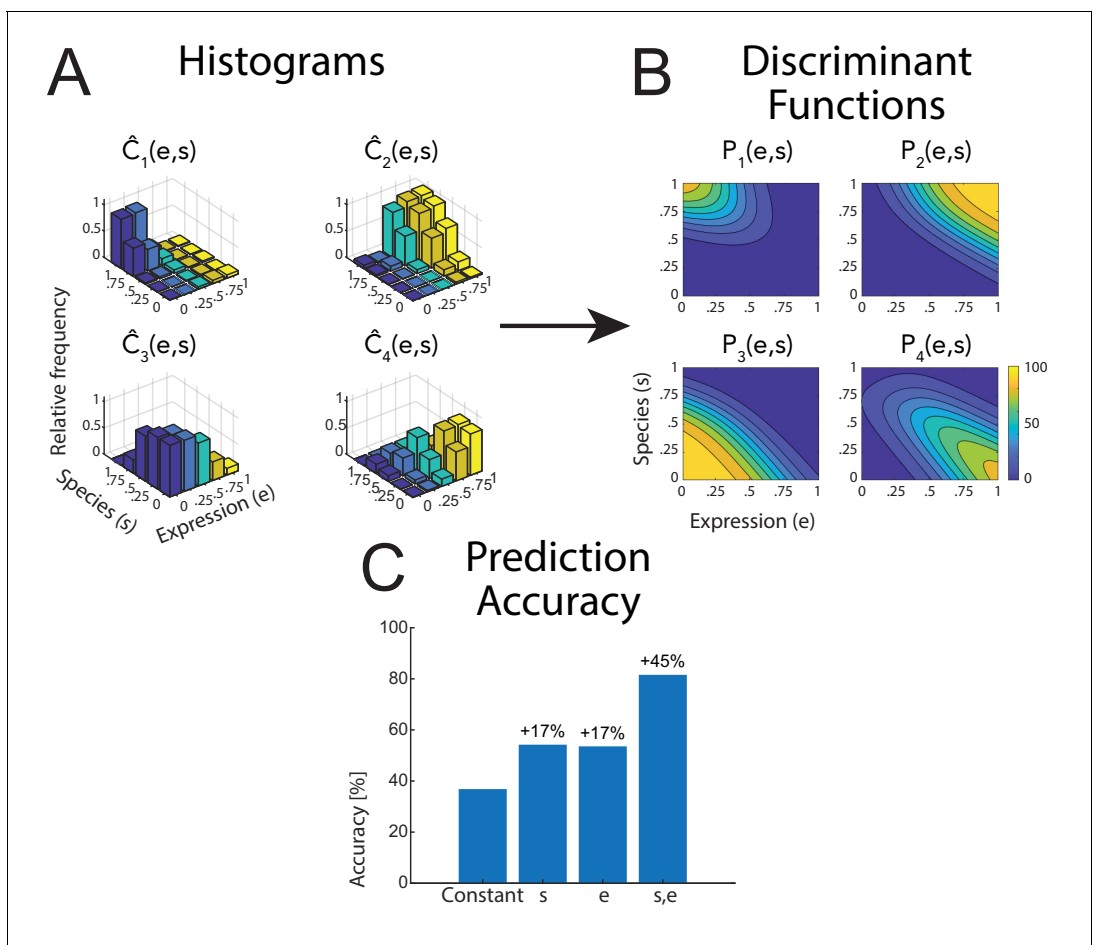

**Figure 2.** Raw data and statistical analysis. (**A**) Histograms of the classification data for the four classes (see text) as functions of the style parameters e and s. Data is shown for the human avatar, front view, using the original motion-captured expressions as prototypes. (**B**) Fitted discriminant functions using a logistic multinomial regression model (see 'Materials and methods'). Data is shown for the human avatar, front view, using the original motion-captured expressions as prototypes. (**C**) Prediction accuracy of the multinomial regression models with different numbers of predictors (constant predictor, only style variable e or s, and both of them).

with different sets of predictors for the monkey avatar stimulus (data from the other conditions are presented in Appendix 1). The different models were compared quantitatively using prediction accuracy and the Bayesian Information Criterion (BIC). Specifically, a model that also included the product $e \cdot s$ did not provide significantly better prediction results, except for a very small improvement of the prediction accuracy for the rotated view conditions. Models only including one of the predictors, $e$ or $s$, provided significantly worse fits. Likewise, models that contained the average amount of optic flow as the additional predictor did not result in higher prediction accuracy (see Appendix 1 for details.). This implies that simple motion features, such as the amount of optic flow, do not provide a trivial explanation of our results. Summarizing, both style variables $e$ and $s$ are necessary as predictors, and there is no strong interaction between them. This motivated us to use the model with these two predictor variables for our further analyses.

*Figure 3A* shows a comparison of all fitted discriminant functions, shown separately for the two avatar types and for the two tested view conditions. These functions show the predicted class probabilities as functions of the two style parameters $e$ and $s$. The form of these discriminant functions is highly similar between the two avatar types and also between the view conditions. This is confirmed by the fact that the fraction of the variance that is different between these functions divided by the one that is shared between them does not exceed 3% ($q = 2.75\%$; see 'Materials and methods'). The same conclusion is also supported by a comparison of the multinomially distributed classification responses using a contingency table analysis (see 'Materials and methods'), across the four conditions (avatar types and views), separately for the different points in morphing space and across participants. This analysis revealed that only for three stimuli (12%) of the style space, the classification responses were significantly different ($p = 0.02$, Bonferroni-corrected). Differences tended to be larger especially for intermediate values of the style space coordinates $e$ and $s$, thus for the stimuli

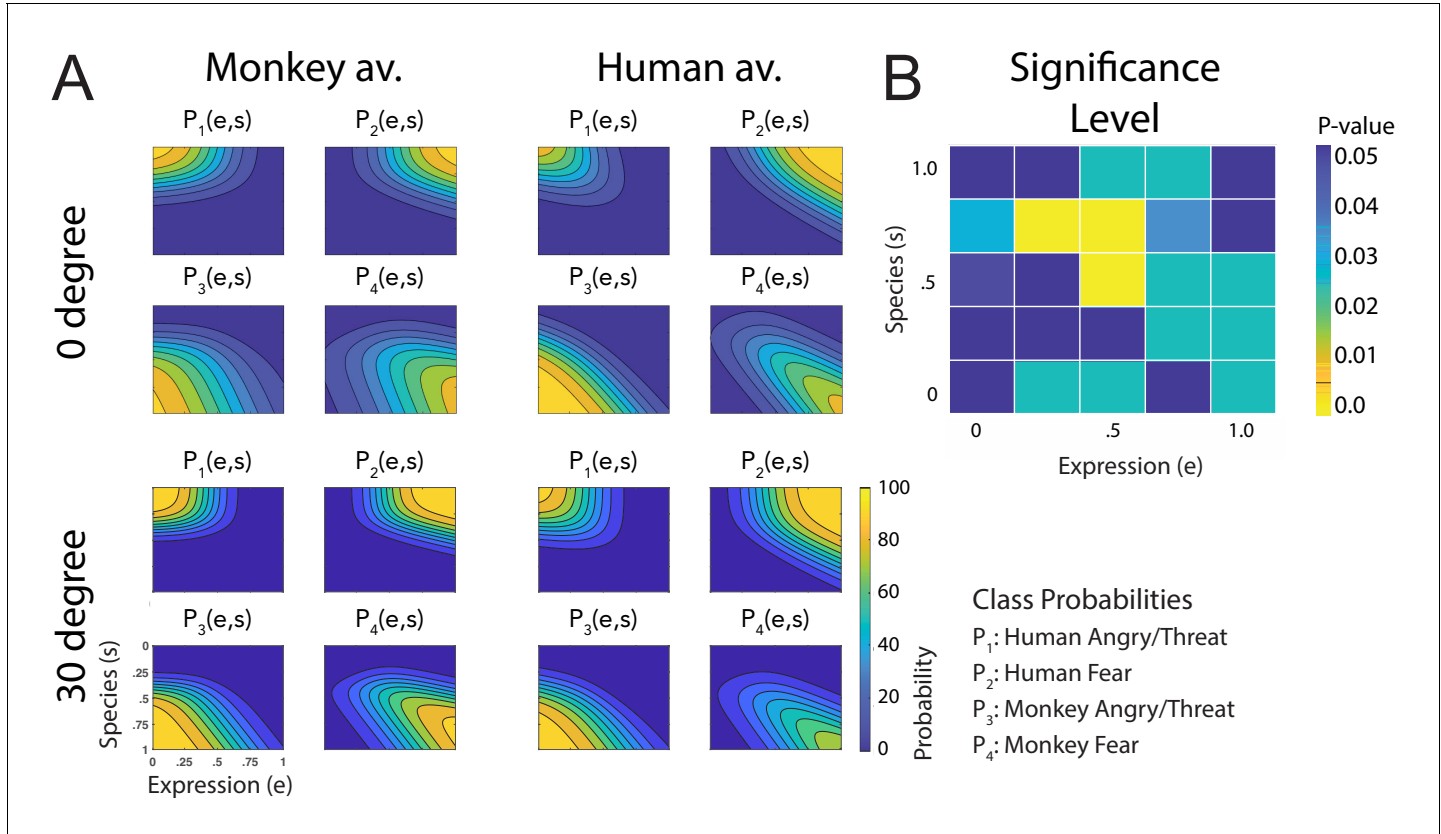

**Figure 3.** Fitted discriminant functions $P_i(e,s)$ for the original stimuli. Classes correspond to the four prototype motions, as specified in *Figure 1D* (i = 1: human-angry, 2: human-fear, 3: monkey-threat, 4: monkey-fear). (**A**) Results for the stimuli generated using original motion-captured expressions of humans and monkeys as prototypes, for presentation on a monkey and a human avatar. (**B**) Significance levels (Bonferroni-corrected) of the differences between the multinomially distributed classification responses for the 25 motion patterns, presented on the monkey and human avatar.

with high perceptual ambiguity (*Figure 3B*). This result implies that primate facial expressions are perceptually encoded largely independently of the head shape (human vs. monkey) and of the stimulus view. Especially, this implies substantial independence of this encoding of the two-dimensional image features, which vary substantially between the view conditions, and even more between the human and the monkey avatar model. This observed independence might also explain why many of our subjects were able to recognize *human* facial expressions on the monkey avatar face, even without any familiarization. This matches the common experience that humans can recognize dynamic facial expressions spontaneously even from non-human comic figures, which often are highly unnatural.

## Tuning is narrower for human-specific than for monkey-specific dynamic expressions

A biologically important question is whether expressions of one's own species are processed differently from those of other primate species, potentially supporting an *own-species advantage* in the processing of dynamic facial expressions (*Dahl et al., 2014*). In order to characterize the tuning of the perceptual representation for monkey vs. human expressions, we computed tuning functions, by marginalizing the discriminant functions belonging to the same species category ($P_1$ and $P_2$ belonging to the human, and $P_3$ and $P_4$ to the monkey expressions) over the expression dimension $e$ (see 'Materials and methods' for details). *Figure 4A* shows the resulting two species-tuning functions $D_H(s)$ and $D_M(s)$, revealing a smaller tuning width for the human than for the monkey expressions for all stimulus types, except for the 30 degrees rotated human condition.

The fitted threshold values are given by the conditions $D_M(s_{th}), D_H(s_{th}) = 0.5$ and are shown in *Figure 4B* for the monkey- and the human-specific motion (solid vs. dashed lines). This observation is confirmed by computing the threshold values of the tuning functions by fitting them with a sigmoidal function (see 'Materials and methods'). Comparing the threshold values by running separate ANOVAs for the four stimulus types (monkey and human front view, and monkey and human rotated view), we found significantly narrower tuning for the human than for the monkey expression for all tested conditions, except for the human avatar in the 30 degrees condition. These two-way mixed-model ANOVAs include the expression type (human vs. monkey motion) as within-subject factor and the stimulus type (original motion, stimuli with occluded ears, or animated with equilibrated motion; see below) as between-subject factor. The ANOVAs reveal a strong effect of the expression type ($F(1, 66) = 188.82$, $F(1, 66) = 46.39$, and $F(1, 40) = 127.35$; $p<0.001$, respectively), except for the human 30 degrees condition, where the influence of this factor did not reach significance ($F(1, 40) = 1.43$; $p>0.23$). In all cases, we failed to find a significant influence of the stimulus type ($F(2, 66) = 0.0$, $F(2, 66) = 0.01$, $F(1, 40) = 0.002$, and $F(1, 40) = 0.014$; $p>0.91$, respectively). Interactions between stimulus type and expression type were found for all conditions ($F(2, 66) = 4.51$; $p<0.015$, $F(2, 66) = 3.15$; $p = 0.049$, $(1, 40) = 8.31$; $p<0.007$, respectively), but not for the human 30 degrees condition ($F(1, 40) = 0.735$; $p>0.39$).

Summarizing, there is a strong tendency of the species-specific expression tuning to be narrower for the human 'own-species' expressions, while this tendency is not as prominent in rotated views.

## Robustness of results against variations of species-specific features

One may ask whether the previous observations are robust with respect to variations of the chosen stimuli. For example, monkey facial movements include *species-specific features*, such as ear motion, that are not present in human expressions. Do the observed differences between the recognition of human and monkey expressions depend on these features? We investigated this question by repeating the original experiment, presenting only the front view, with a new set of participants, using stimuli for which the ear region was occluded. *Figure 5A* depicts the corresponding fitted discriminant functions, which are quite similar to the ones without occlusion, characterized again by a high similarity in shape between the human and monkey avatar (ratio of different vs. shared variance: $q = 1.44\%$; only 12% of the categorization responses over the 25 points in morphing space were significantly different between the two avatar types; $p = 0.02$; *Figure 5B*). *Figure 4A* also shows that the corresponding tuning functions $D_M$ and $D_H$ are very similar to the ones for the non-occluded stimuli, and the associated threshold values (*Figure 4B*) are not significantly different from the one for non-

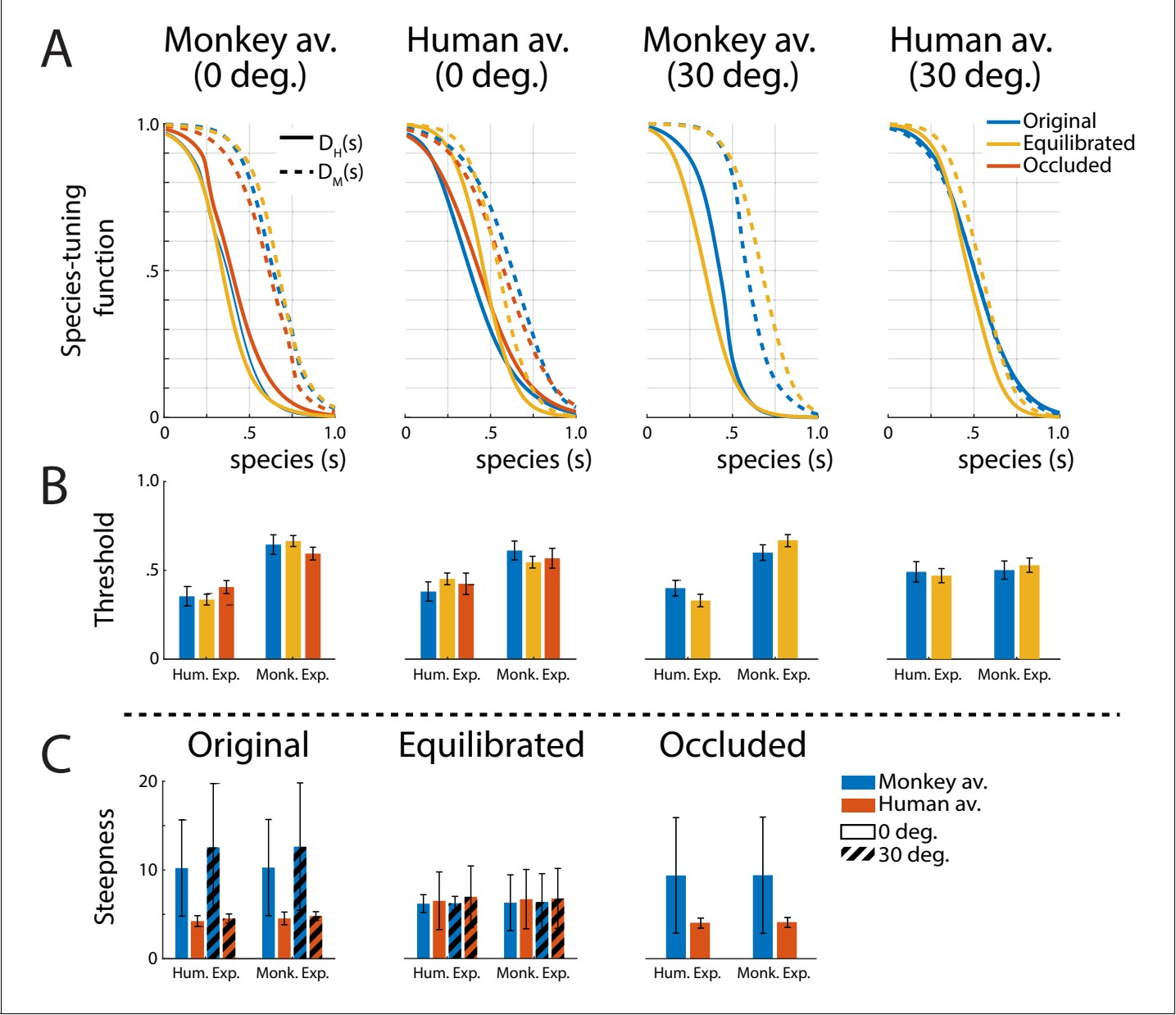

**Figure 4.** Tuning functions. (**A**) Fitted species-tuning functions $D_H(s)$ (solid lines) and $D_M(s)$ (dashed lines) for the categorization of patterns as monkey vs. human expressions, separately for the two avatar types (human and monkey) and the two view conditions. Different line styles indicate the experiments using original motion-captured motion, stimuli with occluded ears, and the experiment using prototype motions that were equilibrated for the amount of motion/deformation across prototypes. (**B**) Thresholds of the tuning functions for the three experiments for presentation on the two avatar types and the two view angles. (**C**) Steepness of the tuning functions at the threshold points for the experiments with and without equilibration of the prototype motions, and with occlusion of the ears.

occluded stimuli (see 'ANOVA analysis'). Summarizing, the elimination of ear motion as a monkey-specific feature did not have a major influence on the main results of the original experiment.

## Robustness against variations of expressivity

A further possible concern might be that the chosen prototypical expressions might specify different amounts of salient low-level features, for example, due to species differences in the motion, or because of differences between the anatomies of the human and the monkey face. In order to control for the influence of such expressive low-level information, we re-ran the main conditions of the

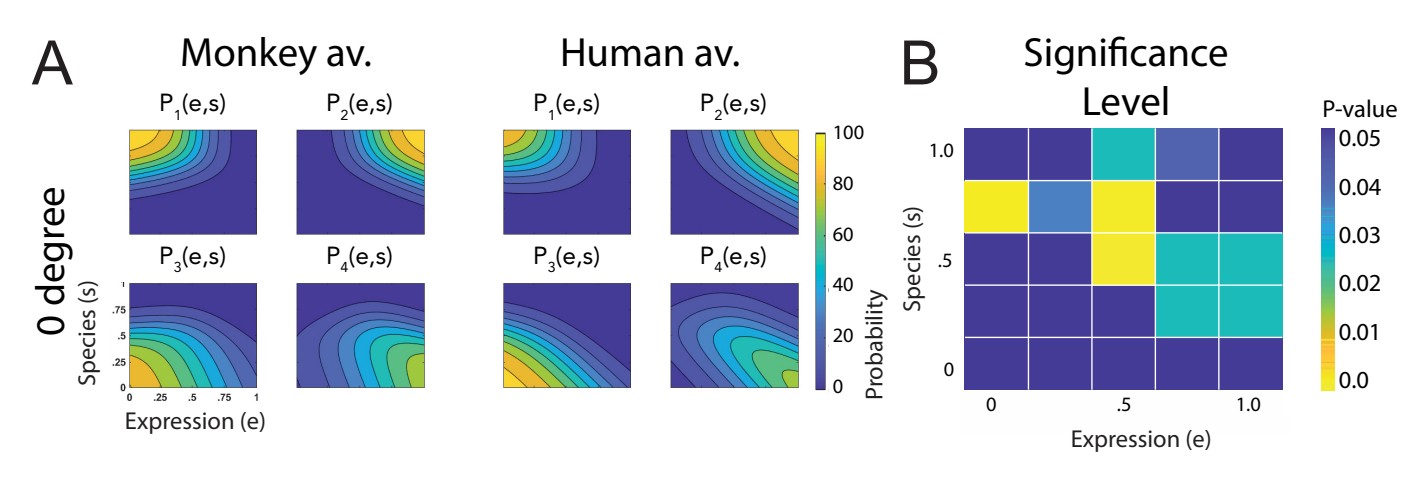

**Figure 5.** Fitted discriminant functions $P_i(e,s)$ for the condition with occlusions of the ears. Classes correspond to the four prototype motions, as specified in *Figure 1D* (i = 1: human-angry, 2: human-fear, 3: monkey-threat, 4: monkey-fear). (**A**) Results for the stimuli generated using original motion-captured expressions of humans and monkeys as prototypes but with occluded ears, for presentation on a monkey and a human avatar (only using the front view). (**B**) Significance levels (Bonferroni-corrected) of the differences between the multinomially distributed classification responses for the 25 motion patterns, presented on the monkey and human avatar.

experiment with stimuli that were equilibrated (balanced) for the amount of such expressive information.

Our equilibration procedure was based on a pilot experiment that compared equilibration methods based on different types of measures for low-level information. This included the total amount of optic flow (OF), the maximum deformation of the polygon mesh during the expression (DF), and the total motion flow of the polygon mesh during the expression (MF) (see 'Materials and methods' for details). In the control experiment, nine participants rated these equilibrated stimulus sets in terms of the perceived expressivity of their motion (independent of avatar type). Perceived expressivity was assessed by ratings using a nine-point Likert scale (1: non-expressive, 9: very expressive), presenting each stimulus in a block-randomized manner for four times.

The averages of these ratings, comparing the different low-level measures, are shown in *Figure 6A*. In addition, this figure also shows the ratings for the neutral expression, which are very low, and the ratings for the original non-equilibrated expressions. It turns out that balancing the amount of polygon motion (MF) resulted in the lowest standard deviation of the expressivity ratings after equilibration (except for the neutral condition $1.479; \mathrm{p}<0.021$).

More specifically, perceived expressivity showed smaller variance for the MF condition than for the DF conditions for the human avatar ($F(1, 142) = 1.479; \mathrm{p}<0.021$). Also, for the monkey avatar, this variance was smaller than for all other conditions ($F>1.403; \mathrm{p}<0.045$), except for the DF condition ($F(1, 142) = 0.869; \mathrm{p}>0.407$). Moreover, the difference of the perceived expressiveness between the two avatars was non-significant ($t(283) = 0.937; p>0.349$) for equilibration with the DF measure. For these reasons, and also because it resulted in the equilibrated stimuli with the highest expressivity, we decided to use MF as a measure of the equilibration of the prototype motion in our main experiment (a more extensive analysis of these data and additional tested measures for low-level expressive information are discussed in Appendix 1).

Equilibration was based on creating morphs between the original motion-captured expressions and a neutral expression, varying the morphing weight of the neutral expression in a way that resulted in a matching of the amount of motion flow (see 'Materials and methods'). Equilibration was realized separately for the two avatars and also for the different view conditions. *Figure 6B* shows an example of the effect of equilibration on the extreme frames of a monkey-threat expression. The equilibration also reduces the very salient mouth opening motion of the monkey, which, due to anatomical differences, cannot be realized by a real human face. The efficiency of the procedure in balancing the amount of motion information is illustrated in *Figure 6C*. It illustrates the motion flow before and after equilibration for the different points of our motion style space for the front view.

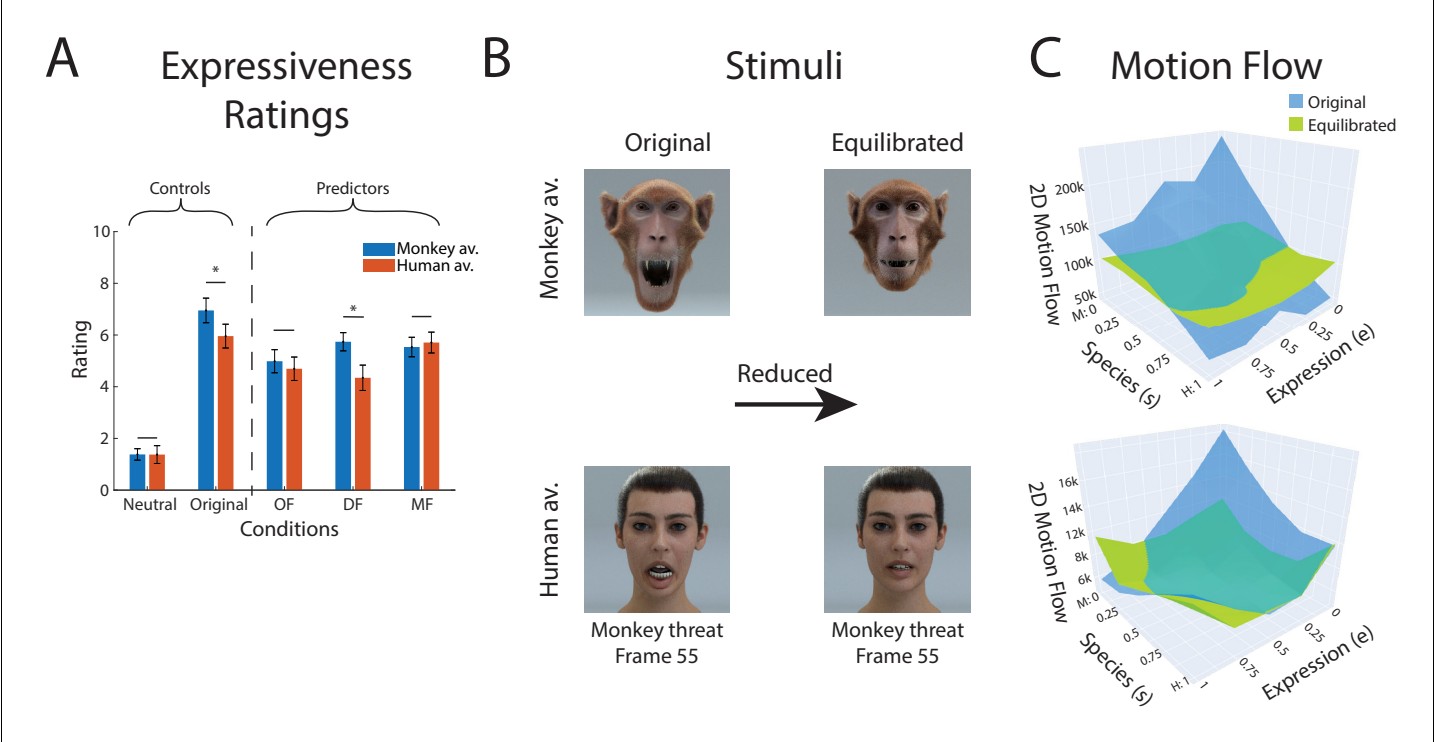

**Figure 6.** Equilibration of low-level expressive information. (**A**) Mean perceived expressivity ratings for stimulus sets that were equilibrated using different types of measures for the amount of expressive low-level information: OF: optic flow computed with an optic flow algorithm; DF: shape difference compared to the neutral face (measured by the 2D distance in polygon vertex space); MF: two-dimensional motion of the polygons on the surface of the face. In addition, the ratings for a static neutral face are shown as reference point for the rating (neutral). (**B**) Extreme frames of the monkey threat prototype before and after equilibration using the MF measure (**C**) 2D polygon motion flow (MF) computed for the 25 stimuli in our expression style space for the monkey avatar for the front view (similar results were obtained for the other stimulus types).

The standard deviation of the motion flow across the 25 conditions in style space is reduced by 83% for the monkey avatar and by 54% for the human avatar by the equilibration. Constraining the flow analysis to the mouth region, we found that the standard deviation of the corresponding motion flow across conditions was reduced by 79% for the monkey avatar and by 59% for the human avatar (results for the other view conditions are similar).

The fitted discriminant functions for the data from the repetition of the experiment with equilibrated stimuli are shown in *Figure 7A*. These functions are more symmetrical along the axes of the morphing space than for the original stimuli (for example, this reduces the amount of confusions of human anger and monkey fear expressions that occurs for intermediate levels of the style parameters, especially for the human avatar, potentially due to the subtlety of the monkey fear expression). This is corroborated by the fact that an asymmetry index (AI) that measures the deviation from a perfect symmetry with respect to the e and s axes (see Appendix 1) is significantly reduced for the data from the experiment with equilibrated stimuli compared to the data from the experiment using the original motion prototypes ($AI_{original} = 0.624$ vs. $AI_{equilibrated} = 0.504$), the difference being significant according to the Wilcoxon signed-rank test ($Z = 2.49; p<0.013$). Compared to the original stimuli, we found an even higher similarity of the discriminant functions between the two avatar types and the different view conditions. This is corroborated by the small ratios of different vs. shared variance between the conditions ($q = 4.01\%$), where only 4% of the categorization responses across the 25 points in morphing space were significantly different between the avatar types and view conditions, according to a contingency table analysis (*Figure 7B*).

Most importantly, also for these equilibrated stimulus sets, we found a narrower tuning for the human than for the monkey dynamic expressions (*Figure 4A*). This is confirmed by the results of the ANOVA for the threshold points of the tuning functions $D_M(s)$ and $D_H(s)$ (*Figure 4B*), which failed to

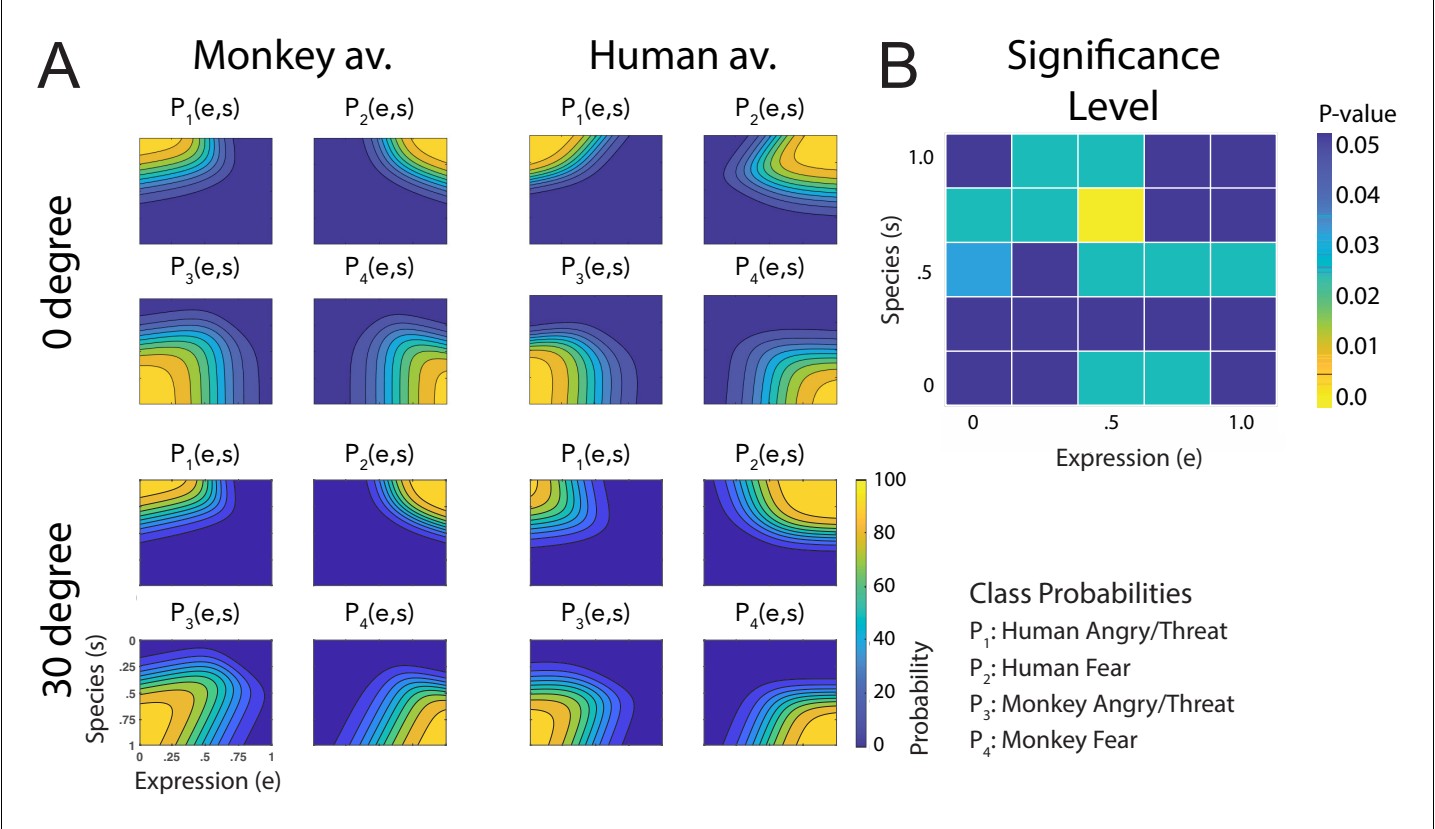

**Figure 7.** Fitted discriminant functions $P_i(e,s)$ for the experiment with equilibration of expressive information. Classes correspond to the four prototype motions, as specified in *Figure 1D* (i = 1: human-angry, 2: human-fear, 3: monkey-threat, 4: monkey-fear). (**A**) Results for the stimuli set derived from prototype motions that were equilibrated with respect to the amount of local motion/deformation information, for presentation on a monkey and a human avatar. (**B**) Significance levels (Bonferroni-corrected) of the differences between the multinomially distributed classification responses for the 25 motion patterns, presented on the monkey and human avatar.

show a significant influence of the factor stimulus type (original vs. occluded vs. equilibrated stimuli) (see above).

An analysis of the steepness of the fitted threshold functions is shown in *Figure 4C*. This analysis shows that the equilibration procedure effectively balances the steepness of the tuning functions between the human and the monkey expressions, which is apparent in the non-equilibrated stimuli. This observation is confirmed by two-way ANOVAs for the original motion stimuli and the ones with occluded ears, which show significant influences of the factor avatar type/view ($F(3,83) = 12.76; p<0.006$, and $F(1,39) = 3.33; p<0.077$, respectively), but not of the expression type ($F(3,83) = 0.01$ and $F(1,39) = 0.01; p>0.92$), and no interactions. Contrasting with this result, the ANOVA for the stimuli with equilibrated motion does not show any significant effects, neither of the avatar type and view ($F(3,87) = 1.27; p>0.26$), nor of the expression type ($F(3,87) = 0.03; p>0.86$), nor of an interaction (full ANOVAS' results in *Appendix 1—table 3*).

Summarizing, these results show that the high similarity of the classification data of the stimuli between the two different avatar types, and between the different view conditions, was not fundamentally changing if the expressiveness of the stimuli was controlled. Also, the tendency for a narrower tuning for human own-species expressions was robust against this manipulation. However, balancing expressiveness leveled out the differences in the steepness of the computed species-tuning functions. This rules out the objection that the observed effects are just an implication of differences in the amount of low-level salient features of the chosen prototypical motion patterns.

# Discussion

Due to the technical difficulties of an exact control of dynamics of facial expressions (*Knappmeyer et al., 2003*; *Hill et al., 2005*), in particular of animals, the computational principles of the perceptual representation of dynamic facial expressions remain largely unknown. Exploiting advanced methods from computer animation with motion capture across species and machine-learning methods for motion interpolation, our study reveals fundamental insights about the perceptual encoding of dynamic facial expressions across primate species. At the same time, the developed technology lays the ground for physiological studies with highly controlled stimuli on the neural encoding of such dynamic patterns (*Polosecki et al., 2013*; *Chandrasekaran et al., 2013*; *Barraclough et al., 2005*; *Furl et al., 2012*).

Our first key observation was that facial expressions of macaque monkeys were learned very quickly by human observers. This was the case even though monkey expressions are quite different from human expressions, so that naive observers could not interpret them spontaneously. This fast learning might be a consequence of the high similarity of the neuromuscular control of facial movements in humans and macaques (*Parr et al., 2010*), resulting in a high similarity of the structural properties of the expression dynamics that can be exploited by the visual system for fast learning.

Secondly, and unexpectedly from shape-based accounts for dynamic expression recognition, we found that the categorization of dynamic facial expressions was astonishingly independent of the type of primate face (human vs. monkey) and of the stimulus view (0 vs. 30 degrees of rotation of the head about the vertical axis). Clearly, this shows a substantial degree of invariance against changes of the two-dimensional image features. More specifically, we neither found strong differences between categorization responses dependent of these parameters, nor did we find a better perceptual representation of species-specific dynamic expressions that match the species of the avatar (e.g., a more accurate representation of human expressions on the human avatar or of monkey expressions on the monkey avatar). Facial expression dynamics seems thus represented independently of the detailed shape features of the primate head and of the stimulus view.

Yet, we found a clear and highly robust own-species advantage (*Scott and Fava, 2013*; *Pascalis et al., 2005*) in terms of the accuracy of the tuning for expression dynamics: the tuning along the species axis of our motion style space was narrower for human than for monkey expressions. This remained true even for stimuli that eliminated species-specific features, such as ear motion, or which were carefully equilibrated in terms of the amount of low-level information.

Both key results support our initial hypotheses: perception can exploit the similarity of the structure of dynamic expressions across different primate species for fast learning. At the same time, and consistent with a co-evolution of the visual processing of dynamic facial expressions with their motor control, we found a largely independent encoding of facial expression dynamics from a basic facial shape in primate expressions. This observed independence has to be further confirmed in more extended experiments, including a bigger spectrum of facial shapes and, probably, even faces from non-primate species. In fact, the observation that humans observe facial expressions readily from comic characters, which do not even correspond to existing species, suggests that the observed invariance goes far beyond primate faces. However, further experiments including a much wider spectrum of facial shapes will be required to confirm this more general hypothesis.

The observed independence of basic facial shape and expression encoding seems in-line with results from functional imaging studies that suggest a modular representation of different aspects of faces, such as changeable and non-changeable ones (*Haxby et al., 2000*; *Bernstein and Yovel, 2015*; *Dobs et al., 2019*). At the same time, it is difficult to reconcile our experiments with several popular (recurrent) neural network models that represent facial expressions in terms of sequences of learned key shapes (*Curio et al., 2010*; *Li and Deng, 2020*). Since the shape differences between human and monkey faces are much larger than the ones between the keyframes from the same expression, the observed spontaneous generalization of dynamic expressions to faces from a different primate species seems difficult to account for by such models.

Concrete circuits for a shape-independent encoding of expression dynamics still have to be discovered. One possibility is that they might exploit optic-flow analysis (*Giese and Poggio, 2003*; *Jhuang et al., 2007*), since the optic flow of the expressions on different head models might be similar. Another possibility would be mechanisms that are based on 'vectorized encoding', where the face shape in individual stimulus frames is encoded in terms of their differences in feature space

from a 'reference' or 'norm face' (*Giese, 2016*; *Leopold et al., 2006*; *Rhodes and Jeffery, 2006*; *Beymer and Poggio, 1996*). We have demonstrated elsewhere that a very robust recognition of dynamic facial expressions can be accomplished by a neural recognition model that is based on this encoding principle (*Stettler, 2020*), where norm-referenced encoding had been shown to account for the identity tuning of face-selective neurons in area IT (*Leopold et al., 2006*; *Giese and Leopold, 2005*). The presented novel technology for the generation of highly realistic dynamic face avatars of humans and monkeys enables electrophysiological studies that clarify the exact underlying neural mechanisms. A similar methodological approach was quite successful for discovering of the neural mechanisms of the identity of static faces (e.g., *Leopold et al., 2006*; *Murphy and Leopold, 2019*; *Chang and Tsao, 2017*).

# Materials and methods

## Key resources table

| Reagent type (species) or resource | Designation | Source or reference | Identifiers | Additional information |
|---|---|---|---|---|
| Software, algorithm | Custom-written software written in C# | This study | https://hih-git.neurologie.uni-tuebingen.de/ntaubert/FacialExpressions (copy archived at swh:1:rev:6d041a0a0cc7055618f85891b85d76e0e7f80eed; *Taubert, 2021*) | |
| Software, algorithm | C3Dserver | Website | https://www.c3dserver.com | |
| Software, algorithm | Visual C++ Redistributable for Visual Studio 2012 Update 4 × 86 and x64 | Website | https://www.microsoft.com/en-US/download/details.aspx?id=30679 | |
| Software, algorithm | AssimpNet | Website | https://www.nuget.org/packages/AssimpNet | |
| Software, algorithm | Autodesk Maya 2018 | Website | https://www.autodesk.com/education/free-software/maya | |
| Software, algorithm | MATLAB 2019b | Website | https://www.mathworks.com/products/matlab.html | |
| Software, algorithm | Psychophysics toolbox 3.0.15 | Website | http://psychtoolbox.org/ | |
| Software, algorithm | R 3.6 | Website | https://www.r-project.org/ | |
| Other | Training data for interpolation algorithm | This study | https://hih-git.neurologie.uni-tuebingen.de/ntaubert/FacialExpressions/tree/master/Data/MonkeyHumanFaceExpression | |
| Other | Stimuli for experiments | This study | https://hih-git.neurologie.uni-tuebingen.de/ntaubert/FacialExpressions/tree/master/Stimuli | |

## Human participants

In total, 78 human participants (42 females) participated in the psychophysical studies. The age range was 21–53 years (mean 26.2, standard deviation 4.71). All participants had no prior experience with macaque monkeys and normal or to-normal corrected vision. Participants gave written informed consent and were reimbursed with 10 EUR per hour for the experiment. In total, 31 participants (16 females) were taking part in the first experiment using stimuli based on the original motion capture data and the experiment with occlusion of the ears. 22 participants (13 females) took part in the experiment with equilibrated motion of the prototypes. In addition, 16 participants (eight females) took part in a Turing test control experiment (see below), and nine (five females) participants took part in a control experiment to identify features that influence perceived expressiveness of the stimuli. All psychophysical experiments were approved by the Ethics Board of the University Clinic Tübingen and were consistent with the rules of the Declaration of Helsinki.

## Stimulus presentation

Subjects were presented the stimuli watching a computer screen at a distance of 70 cm in a dark room (view angle about 12 degrees), with a resolution of 720 × 720 pixels using *MATLAB* and the *Psychotoolbox (3.0.15)* library for stimulus presentation. Each stimulus was repeated for a maximum of three times before asking for the responses, but participants could skip after the first presentation if they were certain about their responses. Participants were first asked whether the perceived expression was rather from a human or a monkey, and whether it was rather the first or the second expression. Responses were given by key presses. Stimuli for the two different avatar types were presented in different blocks, with 10 repeated blocks per avatar type.

## Dynamic monkey and human head model

For our experiments, we exploited a monkey and a human dynamic face avatar with a very high degree of realism. The monkey head model was derived from a structural magnetic resonance scan of a rhesus monkey (9 years old, male). The surface of the face was modeled by an elastic mesh structure (*Figure 1C*) that imitates the deformations induced by the major face muscles of macaque monkeys (*Parr et al., 2010*). To accomplish a highly realistic appearance, special methods for hair animation and an appropriate modeling of skin reflectance were applied (*Figure 1A*). The human head was based on a scan-based commercial avatar with blend-shape animation, exploiting a multi-channel texture simulation software. Mesh deformations compatible with the human face muscle structure were computed from motion capture data in the same way as for the monkey face. Further technical details about the creation of these head models are described in Appendix 1.

The used dynamic head models achieve state-of-the-art degree of realism for the human head, and to our knowledge, we present the only highly realistic monkey avatar that is animated with motion capture data from real animals used in physiology so far. It has been demonstrated by a recent study of our lab that our dynamic monkey avatar induces behavioral reactions of macaque monkeys that are very similar to ones elicited by real movies, reaching the 'good side' of the uncanny valley (*Siebert et al., 2020*), contrasting with previous studies using avatars in experiments with monkeys (*Chandrasekaran et al., 2013*; *Campbell et al., 2009*; *Steckenfinger and Ghazanfar, 2009*). A related result has been obtained recently for static pictures of monkeys, demonstrating comparable looking times for the avatar and real pictures of monkey expressions, but without expressive motion of the face (*Bilder and Lauhin, 2014*).

## Motion generation and style space

The animation of our avatars was based on motion capture data of two real monkey and human expressions. For motion capture, we used a VICON motion capture system with a marker set of 43 markers that were placed on the face of a monkey and a human participant. Facial expressions were elicited by instructions, or by having the animal interact with an experimenter, respectively. For this study, we exploited multiple repetitions of two human and two monkey expressions (*anger/threat* and *fear)*, and additional trials with neutral expressions. Further details about motion capture and the transfer of the motion to the head models are given in Appendix 1.

In order to control the information content and the expressivity of the dynamic face stimuli, we created motion morphs between these prototypical expressions. For this purpose, we exploited a method that is based on a generative Bayesian model of the trajectories of the control points of the face. This algorithm allows to create linear combinations in space-time between these prototypical motions, controlling smoothly the expressiveness and the style of the created facial motion. We verified in an additional control experiment (Turing test) that animations created with the original motion capture data were indistinguishable from the ones generated with motion trajectories generated with this Bayesian algorithm (reproducing the prototypes by the generative Bayesian model) (see Appendix 1 about details concerning this algorithm and the Turing test experiment).

## Modeling of the classification responses

Using a multinomial logistic regression analysis, the relative frequencies of the four classes $\hat{C}_j(e, s)$ were approximated by class probabilities $P_j(e, s)$ for the four classes that were modeled by a generalized linear model (GLM) of the form

$$P_i(e,s) = \frac{e^{y_i}}{\sum_{j'=1}^{4} e^{y_{j'}}} \qquad (1)$$

The variables $y_j$ were given by linear combinations of predictor variables $X_i$ in the form

$$y_j = \beta_{0j} + \beta_{1j}X_1 + \beta_{2j}X_2 + \ldots + \beta_{Nj}X_N \qquad (2)$$

We compared a multitude of models, including different sets of predictors. The most compact model was linear in the style space variables $e$ and $s$ and was given by the equation

$$y_j = \beta_{0j} + \beta_{1j}e + \beta_{2j}s \qquad (3)$$

We also tested variants of linear models that included the predictor variable $e*s$ and a predictor variable that is proportional to the total amount of optical flow, computed using a Horn-Schunck algorithm (CV Toolbox) from the stimulus movies. The different versions of the model were compared exploiting their prediction accuracy and the BIC. We discarded the models if, after addition of a new predictor, either their accuracy was decreasing or the BIC showed a decrease. Further details of the model fitting procedure are described in Appendix 1.

## Computation of the tuning functions

The species-tuning functions were computed by marginalization of the discriminant functions belonging to the same species category along the variable $e$. The tuning function to monkey expressions as a function of the species parameter $s$ was defined as $D_{\mathrm{M}}(s) = \int_0^1 (P_1(e,s) + P_2(e,s))\mathrm{d}e$. Similarly, the tuning function for human expressions was given by $D_{\mathrm{H}}(s) = \int_0^1 (P_3(e, 1-s) + P_4(e, 1-s))\mathrm{d}e$. For this function, the direction of the $s$-axis was flipped, so that the category center also appears for $s = 0$, just as for the function $D_{\mathrm{M}}(s)$.

## Equilibration of stimuli for amount of motion/deformation

In order to control the amount of expressive low-level information, that is, the total amount of motion or shape deformation, we generated sets of equilibrated stimuli. For this purpose, we first defined different measures for the low-level information content and balanced the stimuli by equilibrating these measures. Tested measures included (*Figure 5A*) optic flow (computed with an optic flow algorithm) (OF), the maximum amount of deformation (projected to the plane) of the polygon mesh relative to the neutral pose (DF), and the (two-dimensional) motion flow of the polygon mesh integrated over time (MF). To control the information content of the stimuli, we generated morphs between the original motion and the trajectories of a neutral expression using our motion morphing technique. In these morphs, the original expression was weighted with the morph level $\lambda$ and the neutral expression with the weight $(1 - \lambda)$. The parameter $\lambda$ was chosen to equate the low-level measures of all four prototypical stimuli, separately for the two avatar models (for the front view). For this purpose, we fitted the relationship between the individual measures $M$ for the low-level information and the morphing parameter $\lambda$ by a logistic function of the form ($a_i$ signifying constants)

$$M(\lambda) = a_0 + a_1/(1 + \exp(a_2\lambda + a_3)) \qquad (4)$$

The inverse of this function was used to determine the values of the morph parameter $\lambda$ that matched the value $M$ of the most expressive prototype motion. The MF measure resulted in the least variability of the perceived expressiveness of the equilibrated stimuli (see 'Results'), and thus was used to equilibrate the stimuli for all experimental conditions.

## Statistical analysis

Statistical analyses were implemented using *MATLAB* and RStudio (3.6.2), using R and the package *lme4* for the mixed models of ANOVA. We used G*Power 1.3 software to compute a prior rough estimate of the minimum required number of participants for medium effect size.

Different GLMs for the modeling of the categorization data were fitted using the *MATLAB Statistics Toolbox*. Models for the discriminant functions, including different sets of predictors, were

compared using a step-wise regression approach. Models of different complexity were compared based on the prediction accuracy and by exploiting the BIC.

Two statistical measures were applied in order to compare the similarity of the categorization responses for the two avatar types. First, we computed the ratio of the different vs. shared variance between the fitted discriminant functions. For this purpose, we first computed the average discriminant function across both the avatar types and the two view conditions, and separately for the different classes (the index $k$ running over the avatar types and view conditions, and $j$ indicating the class number):

$$\bar{P}_j(e,s) = \frac{1}{4}\sum_k P_j^k(e,s) \tag{5}$$

The ratio of the variance that is different and shared between the four conditions (avatars and views) is then given by the expression

$$q = \frac{\sum_k \sum_j \iint_0^1 \left(P_{kj}(e,s) - \bar{P}_j(e,s)\right)^2 \mathrm{d}e\mathrm{d}s}{4\sum_{j'} \iint_0^1 \bar{P}_{j'}(e,s)^2 \mathrm{d}e\mathrm{d}s} \tag{6}$$

This ratio is zero if the discriminant functions across all four conditions are identical.

As second statistical analysis, we compared the multinomially distributed four-class classification responses across the participants for the individual points in morphing space using a contingency table analysis that tested for the independence of the class probabilities from the avatar types and the two view conditions. Statistical differences were evaluated using a $\chi^2$-test and, for cases for which predicted frequencies were lower than 5, we exploited a bootstrapping approach (*Wilson et al., 2020*).

The species-tuning functions, $D_H(s)$ and $D_M(s)$, were fitted by the sigmoidal function $D_{H,M} = (\tanh(\omega(s-\theta))+1)/2$, with the parameter $\theta$ determining the threshold and $\omega$, the steepness. Differences of the tuning parameters $\theta$ were tested using two-factor mixed-model ANOVAs (species-specific of motion (monkey vs. human) as the within-subject factor and experiment (original motion, occlusion of the ears, and equilibrated motion) as the between-subject factor). Differences of the steepness parameters $\omega$ were tested using within-subject two-factor ANOVAs.

## Acknowledgements

Special thanks to Tjeerd Dijkstra for the help with advanced statistical analysis techniques. We thank H and I Bülthoff for helpful comments. This work was supported by HFSP RGP0036/2016 and EC CogIMon H2020 ICT-23-2014/644727,and European Research Council ERC 2019-SYG under EU Horizon 2020 research and innovation programme (grant agreement No. 856495, RELEVANCE),. MG was also supported by BMBF FKZ 01GQ1704 and BMG: SSTeP-KiZ (grant no. ZMWI1-2520DAT700). RS, SS, PD, and PT were supported by a grant from the DFG (TH 425/12–2). Further support by NVIDIA Corp.

## Additional information

### Funding

| Funder | Grant reference number | Author |
| --- | --- | --- |
| Human Frontier Science Program | RGP0036/2016 | Martin A Giese |
| Bundesministerium für Bildung und Forschung | BMBF FKZ 01GQ1704 | Michael Stettler<br>Louisa Sting<br>Martin A Giese |
| Baden-Württemberg Stiftung | NEU007/1 KONSENS-NHE | Nick Taubert<br>Michael Stettler<br>Martin A Giese |

| Deutsche Forschungsge-meinschaft | TH 425/12-2 | Ramona Siebert Silvia Spadacenta Peter Dicke Peter Thier |
| --- | --- | --- |
| Nvidia | | Nick Taubert Michael Stettler Martin A Giese |
| European Research Council | 2019-SyG-RELEVANCE-856495 | Nick Taubert Michael Stettler Martin A Giese |
| EC CogIMon H2020 | ICT-23-2014/644727 | Nick Taubert Martin A Giese |

The funders had no role in study design, data collection and interpretation, or the decision to submit the work for publication.

### Author contributions

Nick Taubert, Conceptualization, Data curation, Supervision, Validation, Methodology, Writing - original draft, Writing - review and editing; Michael Stettler, Data curation, Formal analysis, Supervision, Validation, Methodology, Writing - original draft, Writing - review and editing; Ramona Siebert, Data curation, Methodology, Writing - review and editing; Silvia Spadacenta, Louisa Sting, Data curation, Writing - review and editing; Peter Dicke, Supervision, Writing - review and editing; Peter Thier, Conceptualization, Funding acquisition, Project administration, Writing - review and editing; Martin A Giese, Conceptualization, Supervision, Funding acquisition, Validation, Writing - original draft, Project administration, Writing - review and editing

### Author ORCIDs

Nick Taubert (iD) https://orcid.org/0000-0001-5742-3889
Michael Stettler (iD) https://orcid.org/0000-0002-5913-9547
Ramona Siebert (iD) https://orcid.org/0000-0001-5027-8252
Silvia Spadacenta (iD) https://orcid.org/0000-0002-7426-1022
Peter Thier (iD) https://orcid.org/0000-0001-5909-4222
Martin A Giese (iD) https://orcid.org/0000-0003-1178-2768

### Ethics

Human subjects: All participants participated voluntarily in our study and gave written informed consent about this, and about the use of their data in a scientific study. Data was used in a fully anonymised fashion, nowhere we published data from individual participants. All psychophysical experiments were approved by the Ethics Board of the University Clinic Tübingen (886/2018BO2) and were conducted consistently with the rules of the Declaration of Helsinki.

Animal experimentation: All animal procedures were approved by the local animal care committee (Regierungspräsidium Tübingen, Abteilung Tierschutz, permit number: N4/14) and fully complied with German law and the National Institutes of Health's Guide for the Care and Use of Laboratory Animals.

### Decision letter and Author response

Decision letter https://doi.org/10.7554/eLife.61197.sa1
Author response https://doi.org/10.7554/eLife.61197.sa2

## Additional files

### Supplementary files

- Transparent reporting form

## Data availability

Motion Capture data use to train our Bayesian Algorithm, all the rendered stimuli sequences to reproduce our experiment, as well as the raw participants' answers with the source code to reproduce our figures are available on GitLab https://hih-git.neurologie.uni-tuebingen.de/ntaubert/FacialExpressions (copy archived at https://archive.softwareheritage.org/swh:1:rev:6d041a0a0cc7055618f85891b85d76e0e7f80eed).

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

# Appendix 1

## Supporting Information
### Monkey subject

The monkey facial movements were recorded from a 9-year-old male rhesus monkey (*Macaca mulatta*), born in captivity and pair-housed. The monkey had previously been implanted with an individually adapted titanium head-post to allow head immobilization in unrelated neurophysiological experiments, and it had been trained to climb into a primate chair and to accept head fixation. The animal was in daily contact with other macaque monkeys and human caretaking personnel. The structural model of the monkey's head was derived from a T1-weighted MRI-scan with an isotropic resolution of 1 mm. Motion capture recordings were compatible with the guidelines set by the National Institutes of Health and German national law and were approved by the local committee supervising the handling of experimental animals (Regierungspräsidium Tübingen, Abteilung Tierschutz, permit number N4/14). Human movements were recorded from a 40-year-old male human subject.

### Monkey avatar

The highly realistic monkey avatar was generated exploiting state-of-the-art techniques from computer animation. Such techniques have been applied before for the realization of animation movies for cinemas (*Minty, 2014*). However, monkey avatars of high quality have only very recently been developed for studies on static face perception (*Murphy and Leopold, 2019*), and to our knowledge, this work is the first one exploiting motion capture from monkeys for generating such dynamic avatars. The head model was developed based on the MRI scan of an animal (*Appendix 1—figure 1A*). The scan provides a quite detailed model of the basic shape of the head, but it is characterized by a highly irregular polygon structure, which makes it difficult to control the deformation during animation. In order to obtain a mesh model that can be manipulated more easily, we reduced the number of polygons and created a regularized mesh with clean edge loops (*Appendix 1—figure 1B*). This corrected mesh was adjusted for a neutral pose, and control points were specified that control the mesh deformation during animation. For the regularized mesh, the weighting regions that determine the influences of the individual control points on the mesh could be exactly controlled. This developed 'low-polygon' model is useful for controlling the animation, but it lacks a lot of high-frequency details that are critical for a realistic appearance of the face. In order to add such details, we imported the model with the low polygon number into *Adobe Autodesk Mudbox*, a software that allows, by subdivision, to generate again a highly regular mesh model with a high number of polygons. This model was further refined by a number of specific editing steps, including clay modeling, in order to improve 3D shape details. Using a special tool (*Alpha Brushes*), additional texture details were added, such as wrinkles and pores (*Appendix 1—figure 1C*). To transfer the deformation from the low- to the high-frequency polygon model, we exported displacement maps in *Autodesk Mudbox*, which capture the differences between the low and high polygon-density models.

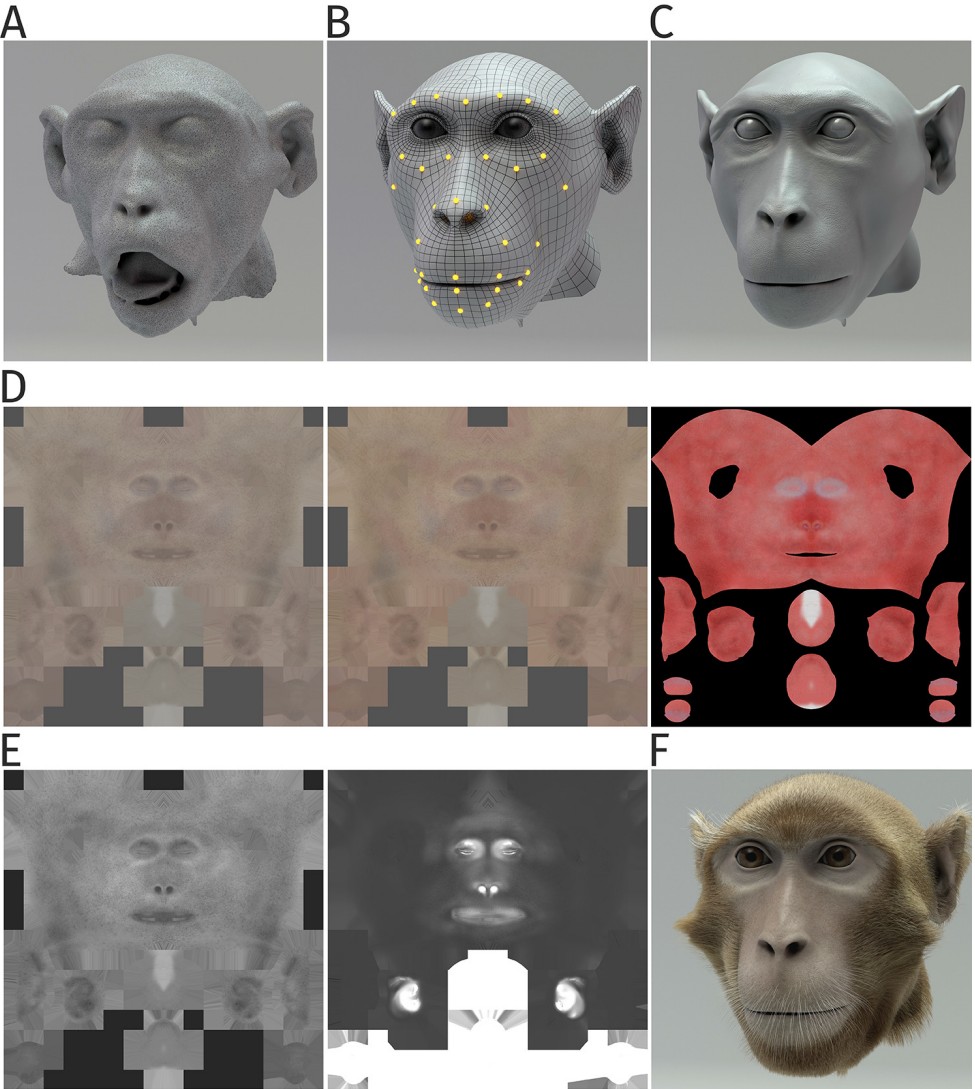

**Appendix 1—figure 1.** Details of generation of the monkey head model. (**A**) Irregular surface mesh resulting from the magnetic resonance scan of a monkey head. (**B**) Face mesh, the deformation of which is following control points specified by motion-captured markers. (**C**) Surface with a high polygon number derived from the mesh in (**B**), applying displacement texture maps, including high-frequency details such as pores and wrinkles. (**D**) Skin texture maps modeling the epidermal layer (left), the dermal layer (middle), and the subdermal layer (right panel). (**E**) Specularity textures modeling the reflection properties of the skin; overall specularity (left) and the map specifying oily vs. wet appearance (right panel). (**F**) Complete monkey face model, including the modeling of fur and whiskers.

A particular challenge was the development of a realistic skin model that specifies believable color and reflectance properties. Skin-surface textures were generated using photographs of a real monkey as the reference and by painting layer-wise color variations of the skin in order to approximate maximally its realistic appearance. Specifically, we used multiple layers of diffusive texture to model the translucent behavior of the skin, separately for the deep layer, subdermal layer, and epidermal layers (*Appendix 1—figure 1D*). For the deep layer, we hue-shifted the diffuse texture map toward red colors in order to model the deep vascularization, while the color of the subdermal layers was shifted more toward yellow in order to simulate the fatty parts of the skin. In order to mimic the very thin superficial layer of dead skin, we desaturated the diffuse texture for the epidermal layer. For realistic appearance, it was also important to model the specularity of monkey skin, reproducing how light is reflected from the skin within different facial regions. For this purpose, we created two

specular maps for the monkey's face, one simulating the basic specularity and one describing the oiliness vs. wetness of the skin (*Appendix 1—figure 1E*). Both material channels have an Index of Refraction (IOR) of 1.375, corresponding to the IOR of water.

A final element that was essential for a realistic appearance was the realistic modeling of the fur. For this purpose, we exploited the built-in *XGen Interactive Groom* feature for hair creation of the animation software *Maya*. The overall appearance of the hair was controlled exploiting three control levels. The first level models the base of the fur, defining the direction of the hairs by control splines and adding some noise to model texture fluctuations and the matting of the fur. The second level models structures consisting of long hair, including the whiskers and the brows, using a smaller number of thick hairs. Believability and realism were increased further by adding a third layer of hair, also known as *Peach Fuzz* or *Vellus*, that consists of tiny hairs that are distributed within the face area. The final result is shown in *Appendix 1—figure 1F*.

## Human avatar

The human avatar was based on a female face scan provided by the company *EISKO* (*Appendix 1—figure 2A*). The commercial package also includes all main textures (diffuse map, specular map, base displacement map, etc; *Appendix 1—figure 2B and C*) for the neutral pose, as well as the corresponding textures for face compression and stretching (*Appendix 1—figure 2B and C*). We applied just small color adjustments to the diffuse and specular maps, similar to texture creation of the monkey head model. The *EISKO* model package also included a whole face rig with 154 blend shapes, suitable for changing the face shape by blending (interpolation), resulting in naturally looking shape variations (*Appendix 1—figure 2A*). The interpolation was driven by control points equivalent to the ones in the monkey model, defined by the motion-captured markers exploiting a ribbon-like structure that was inspired by the human muscle anatomy (*Appendix 1—figure 2F*). Using a tension map algorithm, we determined the local deformations of the texture from the mesh deformations relative to the neutral pose. For the generation of high-frequency details, contrasting with the approach for the monkey avatar, we employed a multichannel texture package from *TexturingXYZ*. This package provides diffuse maps and high-frequency details as displacement maps (pores, wrinkles, etc) derived from a scanned real face. Exploiting the programs *R3dS Wrap* (trial version) and *xNormal*, we transferred shape details similar to the ones of the monkey face to the human face model (*Appendix 1—figure 2G*). Hair animation used the same tools as for the monkey face. The final result is shown in *Appendix 1—figure 2H*.

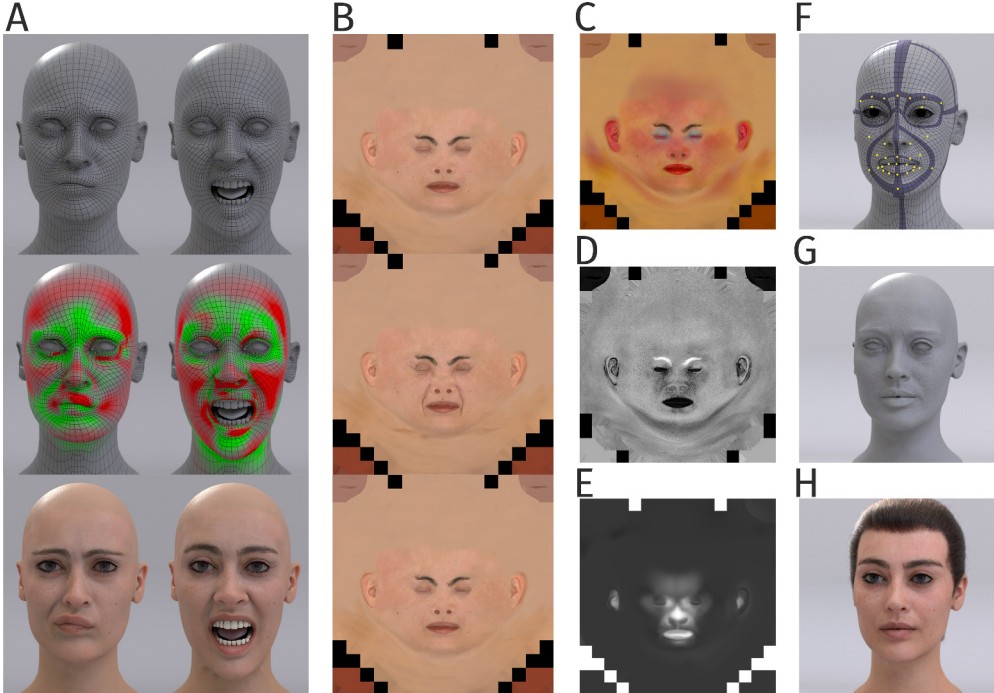

**Appendix 1—figure 2.** Details of generation of the human head model. (**A**) Human face mesh and deformations by a blendshape approach, in which face poses are driven by the 43 control points (top panel). Tension map algorithm computes compression (green) and stretching (red) of mesh during animation (middle panel). Corresponding texture maps were blended (bottom panel). (**B**) Examples of diffuse texture maps (top panel), with additional maps for stretching (middle panel) and compression (bottom panel). (**C**) Subsurface color map modeling the color variations by light scattering and reflection by the skin. (**D**) Specular map modeling the specularity of the skin. (**E**) Wetness map modeling the degree of wetness vs. oilyness of the skin. (**F**) Regularized basic mesh with embedded muscle-like ribbon structures (violet) for animation. Yellow points indicate the control points defined by the motion capture markers. (**G**) Mesh with additional high-frequency details. (**H**) Final human avatar including hair animation.

## Motion capture

Motion capture was realized with a *VICON FX20* motion capture system with six cameras (focal length, 24 mm) using a camera setting that was optimized for face capturing. We used 43 reflecting markers (2 mm) that were placed in the face, using a marker set that we developed ourselves (*Figure 1B* and S1B). Recording frequency was 120 Hz. Trajectories were preprocessed using *Nexus* 1.85 software by *VICON*, smoothed and segmented by an expert into individual facial expressions with a duration between 3 and 5 s. The trajectories were resampled with 150-time steps and 30 fps.

The monkey expressions were recorded from a 9-year-old male animal. The expressions were elicited by showing the animal different objects, including a screw driver, a mirror, and an unknown male human individual. The animal was head-fixed and observed the stimuli at a distance of 200 cm in front of the camera set-up. The recorded trajectories were segmented by a monkey expert, who had extensive experience with macaque monkeys on a daily basis.

The human marker set was corresponding to the one of the monkey, except that it lacked markers on the ears (*Appendix 1—figure 2D*). The human actor was instructed to show two facial expressions 'anger' and 'fear'. Processing was identical to the marker trajectories of the monkey expressions.

## Motion morphing algorithm

In order to create continuous parameterized spaces of facial movements, we exploited a motion morphing method that is based on a hierarchical Gaussian process model. The method is in principle real-time capable, thus allowing for instantaneous changes of motion style modulations based on the on-line user input. This functionality was not critical for the experiments presented in this paper, but it is used in ongoing experiments that build on the presented results.

Our motion morphing algorithm is based on a hierarchical probabilistic generative model that is learned from facial movement data. The architecture (*Appendix 1—figure 3A*) comprises three layers. The lower two layers are formed by Gaussian process latent variable models (GP-LVMs) (*Lawrence and Moore, 2007*), and the highest layer is formed by a Gaussian process dynamical model (GPDM) (*Wang et al., 2008*).

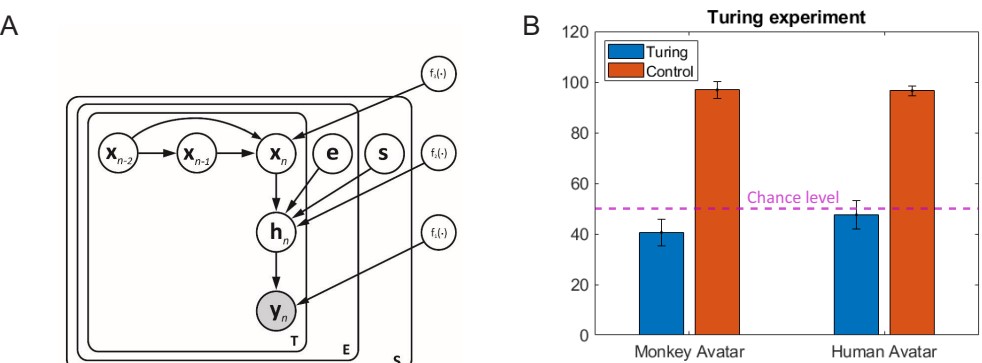

**Appendix 1—figure 3.** Motion morphing algorithm and additional results. (**A**) Graphical model showing the generative model underlying our motion morphing technique. The hierarchical Bayesian model has three layers, reducing subsequently the dimensionality of the motion data $\mathbf{y}_n$. The top layer models the trajectory in latent space using a Gaussian process dynamical model (GPDM). The vectors **e** and **s** are additional style vectors that encode the expression type and the species type. They are binomially distributed. Plate notation indicates the replication of model components for the encoding of the temporal sequence, and the different styles. Nonlinear functions are realized as samples from Gaussian processes with appropriately chosen kernels (for details, see text). (**B**) Results from Turing test experiment. Accuracy for the distinction between animations with original motion capture data and trajectories generated by our motion morphing algorithm is close to chance level (dashed line), opposed to the accuracy for the detection of small motion artifacts in control stimuli, which was almost one for both avatar types.

The facial motion was given by the *M*-dimensional trajectories of the control points, which were parameterized by an *N* x *D* time series matrix $\mathbf{Y} = [\mathbf{y}_1, \ldots, \mathbf{y}_N]^T$ with *N* = 600 (two expressions) or *N* = 900 (neutral expression included for equilibration) and *D* = 208 dimensions. The two GP-LVM layers reduce the dimensionality of the patterns in the high-dimensional trajectory space in a nonlinear way. For this purpose, the first layer represents the trajectory points as nonlinear functions of a lower-dimensional hidden state variable, specifying the *N* x *M* matrix $\mathbf{H} = [\mathbf{h}_1, \ldots, \mathbf{h}_N]^T$. In our case, the dimensionality *M* of the hidden variables $\mathbf{h}_n$ was six. Signifying $(\mathbf{y}^d)^T$, the row vectors of the matrix **Y**, the trajectory components are modeled in the form

$$\mathbf{y}^d = [f_1(\mathbf{h}_n), \ldots, f_1(\mathbf{h}_N)]^T + \varepsilon_d \text{ with } \varepsilon_d \sim \mathcal{N}(\varepsilon_d | 0, \sigma^2 \mathbf{I}),$$

where the variables $\varepsilon_m$ specify independent Gaussian noise vectors, and with the function $f_1$ being drawn from a Gaussian process $f_1 \sim \mathcal{GP}(0, k_1(\mathbf{h}, \mathbf{h}'))$, that is, all vectors of the form $\mathbf{f}_1 = [f_1(\mathbf{h}_n), \ldots, f_1(\mathbf{h}_N)]^T$ are distributed according to the Gaussian distribution $\mathcal{N}(\mathbf{f} | 0, \mathbf{K})$ with the covariance matrix **K**, whose elements are specified by a kernel function $k_1$ in the form $K_{nn'} = k_1(\mathbf{h}_n, \mathbf{h}_{n'}) + \gamma_1 \delta_{nn'}$. The kernel function is given by a linear combination of two types of kernels, a radial basis function kernel and a linear kernel

$$k_1\left(\mathbf{h}, \mathbf{h}'\right) = \gamma_2 \exp\left(-\beta_1 \left|\mathbf{h} - \mathbf{h}'\right|^2\right) + \gamma_3 \mathbf{h}^T \mathbf{h}'$$

The RBF (Radial Basis Function) part allows to capture nonlinear structures in the data, while the linear kernel supports smooth and linear inter- and extrapolation in the pattern space. In addition, we found that the Kronecker delta part of the Kernel matrix is critical for the smoothness of the learned trajectories in the latent space. The parameter $\beta_1$ specifies the inverse width of the Gaussian radial basis functions.

The second layer of the model is defined exactly as the first layer. Here, the dimensionality of the variables $\mathbf{h}_n$ is further reduced by generating a nonlinear mapping from the hidden state variables $\mathbf{x}_n$ with $Q = 2$ dimensions, which defines the matrix $\mathbf{X} = [\mathbf{x}_1, \ldots, \mathbf{x}_N]^\mathrm{T}$. Like in the first layer, the nonlinear mappings between the components of the variables $\mathbf{x}$ and $\mathbf{h}$ are defined by functions drawn from a Gaussian process $f_2 \sim \mathcal{GP}\left(0, k_2\left(\mathbf{x}, \mathbf{s}, \mathbf{e}; \mathbf{x}', \mathbf{s}', \mathbf{e}'\right)\right)$, where the hyper-parameters of the kernel function differ from the ones of the kernel $k_1$. In addition, the kernel of this layer depends on the style vector variables $\mathbf{e}$ and $\mathbf{s}$. These variables enter the kernel of the Gaussian process as multiplicative linear kernel terms

$$k_2\left(\mathbf{x}, \mathbf{s}, \mathbf{e}; \mathbf{x}', \mathbf{s}', \mathbf{e}'\right) = \gamma_4 \mathbf{s}^\mathrm{T} \mathbf{s}' \mathbf{e}^\mathrm{T} \mathbf{e}' \exp\left(-\beta_2 \left|\mathbf{x} - \mathbf{x}'\right|^2\right) + \gamma_5 \mathbf{x}^\mathrm{T} \mathbf{x}'.$$

The random variables $\mathbf{e}$ and $\mathbf{s}$ encode the motion style using one-out-of-two encoding, and they were estimated from the training data together with the state variables $\mathbf{x}_n$ using a maximum-likelihood approach. This parametrization turns out to be favorable to separate the different style components and the motion content in the latent space, similar to multi-factor models (*Wang and Fleet, 2007*). We constrained the style vectors for all trials of the training data that represented the same motion style (e.g., 'expression 1', 'human motion') to be equal. In this way, the training data specify estimates $\hat{\mathbf{e}}_1$ and $\hat{\mathbf{e}}_2$ that correspond to averages of the expression types 1 and 2, and estimates $\hat{\mathbf{s}}_\mathrm{M}$ and $\hat{\mathbf{s}}_\mathrm{H}$ that correspond to the average monkey and the human expressions. In order to generate new intermediate motion styles, we ran the learned Gaussian model in a generative mode, fixing the values of these style vectors to blends between these estimates. The style vectors for the motion morphs as functions of the style parameters $e$ and $s$, as discussed in the main part of the paper, were given by the relationships $\mathbf{e} = e\hat{\mathbf{e}}_1 + (1-e)\hat{\mathbf{e}}_2$ and $\mathbf{s} = s\hat{\mathbf{s}}_\mathrm{M} + (1-s)\hat{\mathbf{s}}_\mathrm{H}$, respectively.

The highest level of the probabilistic model approximates the dynamics of the trajectories of the hidden state variables $\mathbf{x}_n$ using a nonlinear extension of an auto-regressive model, which is known as GPDM. For this purpose, the state dynamics is modeled as a function of the two-dimensional hidden state variable $\mathbf{x}_n$ that obeys the nonlinear dynamics

$$\mathbf{x}_n = f_3(\mathbf{x}_{n-1}, \mathbf{x}_{n-2}) + \xi_n,$$

where $\xi_n$ is isotropic white Gaussian noise and where the nonlinear function $f_3$ is again specified by a Gaussian process. The hidden state dynamics can again be learned using a GP-LVM framework (*Wang et al., 2008*), where we used a kernel function of the form

$$k_3\left(\mathbf{x}_{n-1}, \mathbf{x}_{n-2}; \mathbf{x}_{n'-1}, \mathbf{x}_{n'-2}\right) = \gamma_6 \exp\left(-\beta_3 \left|\mathbf{x}_{n-1} - \mathbf{x}_{n'-1}\right|^2 - \beta_4 \left|\mathbf{x}_{n-2} - \mathbf{x}_{n'-2}\right|^2\right)$$
$$+ \gamma_7 \mathbf{x}_{n-1}^\mathrm{T} \mathbf{x}_{n'-1} + \gamma_8 \mathbf{x}_{n-2}^\mathrm{T} \mathbf{x}_{n'-2}.$$

To determine the parameters of the GP-LVMs, we maximized the logarithm of their marginal likelihood and fitted all hyper-parameters using a scaled conjugate gradient algorithm (*Møller, 1993*). Since the evaluation of the marginal requires the inversion of a kernel matrix with a dimensionality that is given by all pairs of latent points, its direct implementation is computationally infeasible for large data sets. To render this inversion feasible, we applied a sparse approximation method that approximates the marginal distribution based on a low number of inducing points in the hidden spaces (*Lawrence, 2007*). The model was trained using six motion-captured example trajectories for each of the two basic human and monkey expressions, sampled with 150-time steps. Training using an AMD Ryzen Threadripper 1950X CPU with 32 cores with a clock frequency of 3.4 GHz took about 1.5 hr. The most important parameters of the algorithm are summarized in *Appendix 1—table 2*.

## Turing test experiment

The described motion morphing algorithm interpolates between the original motion-captured movements in space-time. It was critical to verify that the morphing algorithm does not destroy the

naturalness of the facial movements, at least for the prototypical expressions between which we blended. In order to verify this question, we realized a Turing test experiment that included 16 new participants. They had to discriminate between animations with original motion capture data ('original trajectories') and ones generated with movements that were generated by the morphing algorithm ('algorithm-generated trajectories'). The movements generated with the algorithm approximated the prototype movements (the style variables $e$ and $s$ being 0 or 1). In order to induce some variability, we used three different motion capture trials of each of the original human and monkey expressions, and their approximations based on the morphing model. The compared stimulus pairs were presented sequentially, and motions were presented in a block-randomized order 20 times, in separate blocks for the two avatar types. To verify that participants can pick up artifacts in the animations, we added a further condition in which instead of movements generated by the morphing algorithm we used control movements, which were generated by reversing the temporal order of short four-frame segments in the original motion-captured movements. Animations with these control movements also had to be distinguished from ones with the original motion capture data.

The results of this control experiments are shown in *Appendix 1—figure 3B*. The accuracy of the detection of original motion capture data, as opposed to the generated one, was 40.6% for the monkey avatar and 47.5% for the human avatar. Compared to the chance probability 0.5, both values were significantly lower ($\chi^2(1, 16) = 18.18; p<0.001$ for the monkey and $\chi^2(1, 16) = 11.43; p<0.001$ for the human avatar). This implies that the animations using motion capture data were judged even less frequently as 'original trajectories' than the animations generated with our motion synthesis algorithm. The morphing algorithm thus does not degrade the perceived naturalness of the motion. The even higher perceived naturalness of the algorithm-generated motion likely is a consequence of the motion being slightly more smooth, due to the smoothing properties of Gaussian process models. The artificial control movements were detected with very high reliability, as indicated by the high accuracies 96.88%, for the monkey, and 96.56%, for the human avatars, which are highly significantly different from chance level ($\chi^2(1, 16) = 35.85; p<0.001$ vs. $\chi^2(1, 16) = 34.93; p<0.001$).

## Comparison of different classification models

Different multinomial regression models were compared in order to find the most compact model that explains our classification data. The models differed in terms of the predictor variables of the linear model for the approximation of the variables $y_j$. The six compared models were defined as

- Model 1: $y_j = \beta_{0j}$.
- Model 2: $y_j = \beta_{0j} + \beta_{1j}e$,
- Model 3: $y_j = \beta_{0j} + \beta_{2j}s$,
- Model 4: $y_j = \beta_{0j} + \beta_{1j}e + \beta_{2j}s$,
- Model 5: $y_j = \beta_{0j} + \beta_{1j}e + \beta_{2j}s + \beta_{3,j}e \cdot s$, and
- Model 6: $y_j = \beta_{0j} + \beta_{1j}e + \beta_{2j}s + \beta_{3,j}OF$.

Apart from the style variables $e$ and $s$, the variable $OF$ signifies the optic flow computed from the image sequence with an optic flow algorithm. Models were compared based on two criteria. First, we required that the introduction of additional predictors did not result in a significantly higher prediction accuracy. According to this criterion, for almost all stimulus types, model four was the most compact model for the front view stimuli (*Appendix 1—table 1*). Only for the rotated views of the avatars, however, we found a slight significant increase of the prediction accuracy (by less than 1.57%). For this reason, we decided to use model four as the basis for our further analyses of all classification data in the main experiment.

## Testing of low-level information that predicts expressivity

Since we found for natural dynamic expressions that a larger part of the tested perceptual space was classified as monkey than as human expressions (*Figure 3C*), we suspected this result to be a potential consequence of monkey expressions specifying more salient low-level features, such as local motion or geometrical deformations. In order to control for this variable, we created a second stimulus set for which the amount of low-level information was balanced. Since it was a priori

unknown which type of low-level information drives the expressivity of facial expressions, we tested nine possible measures, quantifying the amount of low-level features in a separate psychophysical experiment with nine participants. These measures were two-dimensional optic flow computed from the movies with a Horn-Schunck algorithm (*MATLAB* implementation), the absolute spatial deformation relative to the neutral frame, and the motion flow computed either from the control point trajectories or from the regularized mesh points, either in three dimensions or after projection to the two-dimensional image plane.

The *spatial deformation* relative to the neutral frame was quantified using the measures

$$DF = \sum_{t=1}^{N} \|\mathbf{X}_t - \mathbf{X}_0\|_2,$$

where $\mathbf{X}_t$ signifies a vector that contains the relevant control point or (two- or three-dimensional) mesh point coordinates and where $N$ is the number of stimulus frames. Likewise, the *motion flow* was defined by the quantity

$$MF = \sum_{t=2}^{N} \|\mathbf{X}_t - \mathbf{X}_{t-1}\|_2.$$

For the true optic flow, the motion measure was computed by summing up the absolute values of all estimated local motion vectors across the image. The stimuli for this experiment were motion morphs between each of the four prototype expressions (two human and two monkey expressions) and a neutral expression. The original expression entered the motion morph with a weight of $\lambda$, and the neutral expression with a weight of $(1-\lambda)$, where the morphing weight was adjusted to obtain the same amount of low-level information in all adjusted prototypes.

In order to cut down the number of measures for the amount of low-level information in the first place, we generated a set of face motion stimuli with reduced and exaggerated expressivity, separately for the two face avatars, by choosing six different values for the morphing weight $\lambda$ (values 0–25–50–75–100–125% for the monkey expressions and the values 0-37.5–75-112.5–150% for the human expressions). For all rendered movies, we computed the nine different measures for the low-level feature content and analyzed their dependence on the morphing weight $\lambda$ and their similarities. We found that the measures *DF* and *MF,* computed from the two-dimensional and three-dimensional mesh coordinates, and the control points were very highly correlated (r > 86.24, $r_{average}$ = 98.74; p < 0.0403). The mesh point-based measures were monotonically increasing functions of the morphing level $\lambda$. This was not the case for the quantities computed from the control point trajectories, due to which we discarded the measures derived from the control point trajectories from the balancing of the stimuli. Because of the high correlation between the measures computed from the two- and three-dimensional mesh-point trajectories, and the higher similarity of the two-dimensional trajectories with image motion, we kept only the measures computed from the two-dimensional mesh-point trajectories for the further analysis. In addition, we tested the optic flow computed by the optic flow algorithm from pixel images as a third possible predictor of the low-level information. For each of these three predictors, we constructed a balanced stimulus set by adjusting the morph levels of all prototypes, except for the one with the lowest low-level feature content, in order to match their low-level information contents. As a result, we obtained three balanced sets of stimuli, each with four dynamic expressions, separately for each avatar type.

All stimuli were shown in a block-wise randomized order to the participants who had to rate their expressivity on a nine-point Likert scale. For the human avatar, the stimuli, the expressivity of which was balanced using the motion flow measure *MF*, showed the smallest variability across participants and the largest expressivity. For the monkey avatar, the expressivity was rated similarly for stimuli balanced using the measures *MF* and *DF*, while it was significantly lower for stimuli balanced using the optic flow (t(275) = 2.8; p = 0.0054 and t(269) = 3.95; p < 0.001). A step-wise regression analysis, in which we predicted the expressivity ratings from the remaining measures (*MF* and *DF* computed from the two-dimensional mesh motion), showed that the motion flow *MF* is sufficient, while the other predictor *DF* did not add significant additional information. Using a model comparison analysis exploiting the Bayesian Information Criterion (BIC), we found no significant

difference in the explanatory values of the models including the predictor *MF*, and the predictors *MF* and *DF* ( $\chi^2(1, 284) = 3.49; p = 0.062$).

## Asymmetry index

The deviation of the four discriminant functions $P_i(e, s)$ from the completely symmetrical case, where all four discriminant functions have the same basic shape (with their peaks centered on the different prototypes), was quantified by defining the asymmetry index AI. This index is exactly zero if the four discriminant functions are exactly symmetrical with respect to the axes $e = 0.5$ and $s = 0.5$. This implies the symmetry relationship $P_1(e, s) = P_2(1 - e, s) = P_3(e, 1 - s) = P_4(1 - e, 1 - s)$. In order to compute the index, we first computed a symmetrized average of all four discriminant functions according to the formula

$$P_{\text{sym}}(e, s) = \frac{P_1(e, s) + P_2(1 - e, s) + P_3(e, 1 - s) + P_4(1 - e, 1 - s)}{4}$$

Likewise, we defined a standard deviation relative to this symmetrized average by the expression $\text{SD}_{\text{sym}}(e, s) = \sqrt{Q_{\text{sym}}(e, s)/3}$ with the least square deviation sum

$$Q_{\text{sym}}(e, s) = \left(P_1(e, s) - P_{\text{sym}}(e, s)\right)^2 + \left(P_2(1 - e, s) - P_{\text{sym}}(e, s)\right)^2 +$$

$$\left(P_3(e, 1 - s) - P_{\text{sym}}(e, s)\right)^2 + \left(P_4(1 - e, 1 - s) - P_{\text{sym}}(e, s)\right)^2$$

The asymmetry index was defined by the expression

$$\text{AI} = \frac{\iint_0^1 \text{SD}_{\text{sym}}(e, s) \, de \, ds}{\iint_0^1 P_{\text{sym}}(e, s) \, de \, ds}$$

The AI increases with the deviation from the completely symmetric case, where all four categories are represented equally well.

**Appendix 1—table 1.** Model comparison.
Results of the accuracy and the Bayesian Information Criterion (BIC) for the different logistic multinomial regression models for the stimuli derived from the original motion (no occlusions) for the monkey and the human avatar. The models included the following predictors: Model 1: constant; Model 2: constant, *s*; Model 3: constant, *e*; Model 4: constant, *s*, *e*; Model 5: constant, *s*, *e*, product *s·e*; Model 5: constant, *s*, *e*, Optic Flow.

**Model comparison**

| Monkey front view | Model | Accuracy [%] | Accuracy increase [%] | BIC | Parameters | df | $\chi^2$ | p |
|---|---|---|---|---|---|---|---|---|
| | Model 1 | 38.29 | | 7487 | 33 | | | |
| | Model 2 | 57.86 | 19,56 (relative to Model 1) | 5076 | 36 | 3 | 2411 | <0,0001 |
| | Model 3 | 49.49 | 11,2 (relative to Model 1) | 6125 | 36 | 3 | 1362 | <0,0001 |
| | Model 4 | 77.53 | 19,67 (relative to Model 2) | 3586 | 39 | 3 | 1490 | <0,0001 |
| | Model 5 | 77.53 | 0 (relative to Model 4) | 3598 | 42 | 3 | 11.997 | <0.0074 |
| | Model 6 | 77.42 | −0,11 (relative to Model 4) | 3580 | 42 | 3 | 5.675 | 0.129 |
| | | | | | | | | |
| Human front view | | | | | | | | |
| | Model 1 | 36.84 | | 7481 | 33 | | | |
| | Model 2 | 54.22 | 17,38 (relative to Model 1) | 5541 | 36 | 3 | 1940 | <0,0001 |
| | Model 3 | 53.56 | 16,72 (relative to Model 1) | 5847 | 36 | 3 | 1633 | <0,0001 |
| | Model 4 | 81.56 | 27,35 (relative to Model 2) | 3420 | 39 | 3 | 2120 | <0,0001 |

*Continued on next page*

*Appendix 1—table 1 continued*

**Model comparison**

| | | | | | | | | |
|---|---|---|---|---|---|---|---|---|
| | Model 5 | 81.35 | −0,22 (relative to Model 4) | 3309 | 42 | 3 | 112 | <0,0001 |
| | Model 6 | 81.38 | −0,18 (relative to Model 4) | 3389 | 42 | 3 | 31.66 | <0,0001 |

**Monkey 30-degree**

| | | | | | | | | |
|---|---|---|---|---|---|---|---|---|
| | Model 1 | 35.32 | | 6913 | 33 | | | |
| | Model 2 | 57.40 | 22,08 (relative to Model 1) | 4314 | 36 | 3 | 2622 | <0,0001 |
| | Model 3 | 49.36 | 14,04 (relative to Model 1) | 5179 | 36 | 3 | 1757 | <0,0001 |
| | Model 4 | 84.04 | 26,64 (relative to Model 2) | 2359 | 39 | 3 | 1977 | <0,0001 |
| | Model 5 | 84.88 | 0,84 (relative to Model 4) | 2335 | 42 | 3 | 48 | <0,0001 |
| | Model 6 | 84.08 | 0,04 (relative to Model 4) | 2331 | 42 | 3 | 28 | <0,0001 |

**Human 30-degree**

| | | | | | | | | |
|---|---|---|---|---|---|---|---|---|
| | Model 1 | 37.40 | | 6819 | 33 | | | |
| | Model 2 | 55.72 | 18,32 (relative to Model 1) | 4843 | 36 | 3 | 1975 | <0,0001 |
| | Model 3 | 54.36 | 16,96 (relative to Model 1) | 5217 | 36 | 3 | 1602 | <0,0001 |
| | Model 4 | 81.32 | 25,6 (relative to Model 2) | 2910 | 39 | 3 | 1956 | <0,0001 |
| | Model 5 | 82.88 | 1,56 (relative to Model 4) | 2809 | 42 | 3 | 101 | <0,0001 |
| | Model 6 | 81.92 | 0,6 (relative to Model 4) | 2890 | 42 | 3 | 19 | 0.0002 |

**Appendix 1—table 2.** Parameters of the Bayesian motion morphing algorithm.
The observation matrix $Y$ is formed by $N$ samples of dimension $D$, where $N$ results from $S * E$ trails with $T$ time steps. The dimensions $M$ and $Q$ of the latent variables were manually chosen. The integers $S$ and $E$ specify the number of species and expressions (two in our case).

**Parameters of motion morphing algorithm**

| Parameters | Description | Value |
|---|---|---|
| $D$ | Data dimension | 208 |
| $M$ | First layer dimension | 6 |
| $Q$ | Second layer dimension | 2 |
| $T$ | Number of samples per trial | 150 |
| $S$ | Number of species | 2 |
| $E$ | Number of expressions | two or 3 |
| $N$ | Number of all samples | $T * S * E$ |
| **Hyper parameters (learned)** | | **Size** |
| $\beta_1$ | Inverse width of kernel $k_1$ | 1 |
| $\beta_2$ | Inverse width of kernel $k_2$ | 1 |
| $\beta_3$ | Inverse width for non-linear part one of kernel $k_3$ | 1 |
| $\beta_4$ | Inverse width for non-linear part two of kernel $k_3$ | 1 |
| $\gamma_1$ | Precision absorbed from noise term $\varepsilon_d$ | 1 |
| $\gamma_2$ | Variance for non-linear part of $k_1$ | 1 |
| $\gamma_3$ | Variance for linear part of $k_1$ | 1 |
| $\gamma_4$ | Variance for non-linear part of $k_2$ | 1 |
| $\gamma_5$ | Variance for linear part of $k_2$ | 1 |
| $\gamma_6$ | Variance for non-linear part of $k_3$ | 1 |

*Continued on next page*

| $\gamma_7$ | Variance for linear part one of $k_3$ | 1 |
|---|---|---|
| $\gamma_8$ | Variance for linear part two of $k_3$ | 1 |
| **Variables** | | **Size** |
| Y | Data | *N x D* |
| H | Latent variable of first layer | *N x M* |
| X | Latent variable of second layer | *N x Q* |
| $\mathbf{s}_M$ | Style variable vector for monkey species | *S x 1* |
| $\mathbf{s}_H$ | Style variable vector for human species | *S x 1* |
| $\mathbf{e}_1$ | Style variable vector for expression one | *E x 1* |
| $\mathbf{e}_2$ | Style variable vector for expression two | *E x 1* |

**Appendix 1—table 3.** Detailed results of the two-way ANOVAs.

ANOVA for the threshold: two-way mixed model with expression type as within-subject factor and the stimulus type as between-subject factor for both the monkey and the human avatar. Steepness: two-way ANOVA with avatar type and expression factor for each stimulus motion type (original, occluded, and equilibrated). The mean square is defined as Mean Square = Sum of Square/df; df = degree of freedom.

**ANOVAs**

| Threshold | *Monkey avatar* | Sum of square | df | Mean square | F | p |
|---|---|---|---|---|---|---|
| | Stimulus type | 0,00 | 2 | 0,00 | 0,00 | 0999 |
| | Expression type | 1,20 | 1 | 1,20 | 188,83 | 0000 |
| | Stimulus * Expression | 0,06 | 2 | 0,03 | 4,51 | 0015 |
| | Error | 0,42 | 60 | 0,01 | | |
| | Total | 1,72 | 65 | | | |
| | | | | | | |
| | *Human avatar* | | | | | |
| | Stimulus type | 0,00 | 2 | 0,00 | 0,01 | 0993 |
| | Expression type | 0,40 | 1 | 0,40 | 46,37 | 0000 |
| | Stimulus * Expression | 0,05 | 2 | 0,03 | 3,15 | 0049 |
| | Error | 0,57 | 60 | 0,01 | | |
| | Total | 1,02 | 65 | | | |
| | | | | | | |
| Steepness | *Original motion stimulus* | | | | | |
| | Avatar type | 376,68 | 1 | 376,68 | 6,3 | 0016 |
| | Expression type | 0,36 | 1 | 0,36 | 0,01 | 0939 |
| | Avatar * Expression | 0,16 | 1 | 0,16 | 0 | 0959 |
| | Error | 2391,21 | 40 | 59,78 | | |
| | Total | 2768,41 | 43 | | | |
| | | | | | | |
| | *Occluded motion stimulus* | | | | | |
| | Avatar type | 286,17 | 1 | 286,17 | 3,33 | 0076 |
| | Expression type | 0,02 | 1 | 0,02 | 0 | 0988 |
| | Avatar * Expression | 0,00 | 1 | 0,00 | 0 | 0995 |
| | Error | 3094,54 | 36 | 85,96 | | |
| | Total | 3380,73 | 39 | | | |

*Continued on next page*

| | | | | | |
|---|---|---|---|---|---|
| *Equilibrated motion stimulus* | | | | | |
| Avatar type | 1,57 | 1 | 1,57 | 0,4 | 0533 |
| Expression type | 0,25 | 1 | 0,25 | 0,06 | 0803 |
| Avatar * Expression | 0,02 | 1 | 0,02 | 0 | 0945 |
| Error | 174,76 | 44 | 3,97 | | |
| Total | 176,60 | 47 | | | |

