## [Decision Letter]

**Acceptance summary:**

This paper employs novel cross species stimuli (human and monkey) and a well-designed psychophysical paradigm to study the processing of dynamic facial expressions. Strikingly, the study shows that facial expression discrimination is largely independent on whether the expression is conveyed by a human or a monkey face. The novel photo-realistic dynamic avatars developed here will allow future studies to address novel questions about facial expression and social communication.

**Decision letter after peer review:**

Thank you for submitting your article "Shape-invariant perceptual encoding of dynamic facial expressions across species" for consideration by *eLife*. Your article has been reviewed by 2 peer reviewers, and the evaluation has been overseen by a Reviewing Editor and Timothy Behrens as the Senior Editor. The reviewers have opted to remain anonymous.

The reviewers have discussed the reviews with one another and the Reviewing Editor has drafted this decision to help you prepare a revised submission.

Summary:

The paper employs novel cross species stimuli and a well-designed psychophysical paradigm to study the visual processing of facial expression. The authors show that facial expression discrimination is largely invariant to face shape (human vs. monkey). Furthermore, they reveal sharper tuning for recognising human expressions compared to monkey expressions, independent of whether these expressions were conveyed by a human or a monkey face. Technically, the paper is of a very high quality, both in terms of stimulus creation, but also in terms of analysis.

1. A central claim of the paper and the first words in the title are that the behavior studied (categorization of facial expression dynamics) is "shape-invariant". However, the lack of variation in facial shapes (n = 2) used here limits the strength of the conclusions that can be drawn, and it certainly remains an open question whether representations of facial expression dynamics are truly "shape-invariant". A simple control would have been to vary the viewing angle of the avatars, in order to dissociate 3D object shapes from their 2D projections (images). The authors also claim that "face shapes differ considerably" (line 49) amongst primate species, which is clearly true in absolute terms. However, the structural similarity of simian primate facial morphology (i.e. humans and macaques used here) is striking when compared to various non-primate species, which naturally raises questions about just how shape-invariant facial expression recognition is. The lack of data to more thoroughly support the central claim is problematic. In the absence of additional data, the authors should tone down this claim.

2. As the authors note, macaque and human facial expressions of 'fear' and 'threat' differ considerably in visual salience and motion content – both in 3D and their 2D projections (i.e. optic flow). Indeed, the decision to 'match' expressions across species based on semantic meaning rather than physical muscle activations is a central problem here. Figure 1A illustrates clearly the relative subtlety of the human expression compared to the macaque avatar's extreme open-mouthed pose, while Figure 1D (right panels) shows that this is also true of macaque expressions mapped onto the human avatar. The authors purportedly controlled for this in an 'optic-flow equilibrated' experiment that produced similar results. However, this crucial control is currently difficult to assess since the control stimuli are not illustrated and the description of their creation (in the supplementary materials) is rather convoluted and obfuscates what the actual control stimuli were.

The results of this control experiment that are presented (hidden away in supplementary Appendix 3—figure 1C) show that subjects rated the equilibrated stimuli at similar levels of expressiveness for the human vs. macaque avatars. However, what the reader really needs to know is whether subjects rated the human vs. macaque expression dynamics to be similarly expressive (irrespective of avatar)? My understanding is that species expression (and not species face shape) is the variable that the authors were attempting to equilibrate for.

In short, the authors have not presented data to convince a reader that their equilibrated stimuli resolve the obvious confound in their original stimuli (namely the correlation between low level visual salience – especially around the mouth region – and the species of the expression dynamics). These data should either be presented, or the authors' claims should be toned down.

3. This paper appears to be the human psychophysics component of work that the authors have recently published using the macaque avatar. The separate paper (Siebert et al., 2020 – eNeuro) reported basic macaque behavioral responses to similar animations, while the task here takes advantage of the more advanced behavioral methods that are possible in human subjects. Nevertheless, the emphasis of the current paper on cross-species perception raises the question – how do macaques perceive these stimuli? Do the authors have any macaque behavioral data for these stimuli (even if not for the 4AFC task) that could be included to round this out? If not, we recommend rewording the title since its current grammatical structure implies that the encoding is "across species", whereas encoding *of species* (shape and expression) was only tested in one species (humans).

4. The authors may want to consider restructuring the Results section. The main take-home messages do not come through clearly in the way the Results section is currently structured. It contains a lot of technical detail – despite considerable use of Supplementary Information (SI) – which made extracting the empirical findings quite hard work. The details of the multinominal regression, the model comparisons (Table 1) and even the Discriminant Functions (Figure 2), for example, could all be briefly mentioned in the main text, with details provided in Methods or SI. These are all interesting, but the focus of the Results section should be on the behavioural findings, not the methods. The authors could use their Discussion – which clearly states the key points – as a guide, making sure the focus is more on Figure 3 and then working through the points more concisely.

Revisions expected in follow-up work:

1. Future work should expand the set of face shapes in order to properly test for shape invariance.

[Editors' note: further revisions were suggested prior to acceptance, as described below.]

Thank you for resubmitting your work entitled "Shape-invariant encoding of dynamic primate facial expressions in human perception" for further consideration by *eLife*. Your revised article has been reviewed by 2 peer reviewers, and the evaluation has been overseen by a Reviewing Editor and Timothy Behrens as the Senior Editor.

The manuscript has been very much improved. We were particularly impressed with the additional experiments conducted. There are some remaining issues that need to be addressed, as outlined below:

– The additional explanation of the analysis now provided by Figure 2 is welcome, but the layout and labeling of the main result in Figure 3 is still rather taxing on the reader and suboptimal for conveying the main result ('shape invariance'). In fact, Figure 3 is now less focussed as a result of including more data. Readers currently have to memorize which species and expression each of the 4 'style parameters' ("P1(e,s) – P4(e,s)") correspond to, and then parse a grid of 40 such plots in order to discern which they should be visually comparing. The previous iteration of this figure (previously Figure 2) with one column for macaque avatar and another for human avatar made this a bit easier. Minimally, the authors could improve the labeling of plots in Figure 3 to help readers parse the large array of heatmaps.

– The authors continue to insist that their macaque avatar achieves "the best known degree of realism" to date. There does not seem to be empirical evidence to support such a claim, since the authors' previous work (Siebert et al., 2020) did not compare their own avatar to other groups' avatars. At least one other group has found similar looking preference results for their own macaque avatar (Wilson et al., 2020).

– The links provided to online data and stimuli (https://hih-git.neurologie.uni-tuebingen.de/ntaubert/FacialExpressions) appear not to be publicly accessible at present. Will this be changed before publication or does access need to be requested? If the latter, instructions for obtaining access should be provided.

---

## [Author Response]

Revisions for this paper:1. A central claim of the paper and the first words in the title are that the behavior studied (categorization of facial expression dynamics) is "shape-invariant". However, the lack of variation in facial shapes (n = 2) used here limits the strength of the conclusions that can be drawn, and it certainly remains an open question whether representations of facial expression dynamics are truly "shape-invariant". A simple control would have been to vary the viewing angle of the avatars, in order to dissociate 3D object shapes from their 2D projections (images). The authors also claim that "face shapes differ considerably" (line 49) amongst primate species, which is clearly true in absolute terms. However, the structural similarity of simian primate facial morphology (i.e. humans and macaques used here) is striking when compared to various non-primate species, which naturally raises questions about just how shape-invariant facial expression recognition is. The lack of data to more thoroughly support the central claim is problematic. In the absence of additional data, the authors should tone down this claim.

In fact, the possibility to test different head shapes in our study had been strongly limited by the enormous technical effort that is necessary to create such highly believable dynamic avatar models, even for one head shape. Unfortunately, this methodological limitation cannot easily be overcome. We decided thus to follow the reviewers’ suggestion to increase the number of tested stimuli by including view variations. This increases the number of tested 2D facial patterns that are associated with the same dynamic expression. To accommodate this suggestion, we have repeated a substantial part of our experiments with patterns with a different view. This view was taken as dissimilar to the front view as possible, without producing salient occlusions or artifacts. As result, we found that our main hypothesis is largely confirmed, even for this more variable stimulus set.

In order to accommodate this suggested extension, we had to change multiple things in the manuscript:

– We have added the new data from the repetition of two experiments with different stimulus views to the main paper and have adjusted the statistical analysis. As result, the amount of presented data now is substantially larger. This made it necessary to reorganize the figures in the manuscript in order to keep the presentation of the data understandable. Specifically, we have increased the number of figures from 3 to 5. Several figure panels had to be assigned to new figures for this purpose.

– We have toned down and refined the claims about shape invariance in an absolute sense. Specifically, we have introduced a distinction of 3D and 2D shape, and we have restricted some claims to the class of faces of primates. Also we have added a short discussion of the mentioned limitations to the Discussion section.

2. As the authors note, macaque and human facial expressions of 'fear' and 'threat' differ considerably in visual salience and motion content – both in 3D and their 2D projections (i.e. optic flow). Indeed, the decision to 'match' expressions across species based on semantic meaning rather than physical muscle activations is a central problem here. Figure 1A illustrates clearly the relative subtlety of the human expression compared to the macaque avatar's extreme open-mouthed pose, while Figure 1D (right panels) shows that this is also true of macaque expressions mapped onto the human avatar. The authors purportedly controlled for this in an 'optic-flow equilibrated' experiment that produced similar results. However, this crucial control is currently difficult to assess since the control stimuli are not illustrated and the description of their creation (in the supplementary materials) is rather convoluted and obfuscates what the actual control stimuli were.The results of this control experiment that are presented (hidden away in supplementary Figure S3C) show that subjects rated the equilibrated stimuli at similar levels of expressiveness for the human vs macaque avatars. However, what the reader really needs to know is whether subjects rated the human vs macaque expression dynamics to be similarly expressive (irrespective of avatar)? My understanding is that species expression (and not species face shape) is the variable that the authors were attempting to equilibrate for.In short, the authors have not presented data to convince a reader that their equilibrated stimuli resolve the obvious confound in their original stimuli (namely the correlation between low level visual salience – especially around the mouth region – and the species of the expression dynamics). These data should either be presented, or the authors' claims should be toned down.

In fact, our control stimuli were designed to control for the perceived expressiveness of the dynamic expression within the individual avatars, but across different species-specific motion types. However, the finally chosen equilibration method also has the consequence that the expressiveness between the two avatar types was balanced. (Relevant data has been added and is discussed more clearly now.) The reviewers are right that the motivation and the details about the equilibration were difficult to access in the old version of the manuscript. In addition, the description was not always very clear. We have tried to fix these issues and have substantially reorganized the material to adopt this criticism. More specifically, we have made the following changes:

– The description and motivation of the equilibration procedure has been massively rewritten in the main paper as well as in the Supplemental Information. The data supporting the efficiency of the equilibration method has been integrated in the main paper.

– We have tried to improve substantially the clarity of the explanation of the equilibration procedure and of the underlying assumptions. The Supplemental Information has been adopted to this reorganization of the material.

– We have added a new figure (Figure 5) showing examples of the original as well as of the equilibrated expressions for both avatar types. In addition, this figure shows the efficiency of the chosen procedure to balance the motion flow, as well as the expressiveness rating data that justifies the chosen method.

– We have quantified the efficiency of the procedure in terms of equilibrating the motion flow for the whole face, and separately also for the mouth region, finding a quite similar reductions of the variability of low-level information across the different conditions in style space.

3. This paper appears to be the human psychophysics component of work that the authors have recently published using the macaque avatar. The separate paper (Siebert et al., 2020 – eNeuro) reported basic macaque behavioral responses to similar animations, while the task here takes advantage of the more advanced behavioral methods that are possible in human subjects. Nevertheless, the emphasis of the current paper on cross-species perception raises the question – how do macaques perceive these stimuli? Do the authors have any macaque behavioral data for these stimuli (even if not for the 4AFC task) that could be included to round this out? If not, we recommend rewording the title since its current grammatical structure implies that the encoding is "across species", whereas encoding of species (shape and expression) was only tested in one species (humans).

We agree with the reviewers that it would be very attractive to have also macaque psychophysical data on the same stimulus set. Unfortunately, this would imply a major experimental effort because the animals would have to be trained on an active task, requiring a discrimination the expression type independent of the avatar type, without the possibility to give direct instructions about this to the animal. We thus followed the reviewers’ suggestion to reword the title, in order to reflect more appropriately that our study is based on human data only. As new title we chose: ‘Shape-invariant encoding of dynamic primate facial expressions in human perception’. We also have carefully corrected the argumentation in the rest of the manuscript to avoid misunderstandings concerning the mentioned point.

4. The authors may want to consider restructuring the Results section. The main take-home messages do not come through clearly in the way the Results section is currently structured. It contains a lot of technical detail – despite considerable use of Supplementary Information (SI) – which made extracting the empirical findings quite hard work. The details of the multinominal regression, the model comparisons (Table 1) and even the Discriminant Functions (Figure 2), for example, could all be briefly mentioned in the main text, with details provided in Methods or SI. These are all interesting, but the focus of the Results section should be on the behavioural findings, not the methods. The authors could use their Discussion – which clearly states the key points – as a guide, making sure the focus is more on Figure 3 and then working through the points more concisely.

We thank the reviewers and the editors very much for this suggestion. We have massively restructured the paper to separate more clearly the Results and the Methods part. Several methodological details have been shifted to the Methods section, and specifically all equations. We tried, however, to keep the Results understandable by providing the necessary minimal information about the relevant methods. The Methods section has been restructured in order to accommodate the relevant additional material. Very detailed or technical parts of the methods have been reserved for the Supplementary Information.

Revisions expected in follow-up work:1. Future work should expand the set of face shapes in order to properly test for shape invariance.

We have added a substantial amount of new data with different 2D shapes by testing stimuli with a different view angle. We have also added a discussion of this limitation to the final Discussion section, where we point to this limitation and the necessity to test a broader set of head shapes, including also non-primate faces, defining an interesting direction for future work.

[Editors' note: further revisions were suggested prior to acceptance, as described below.]

The manuscript has been very much improved. We were particularly impressed with the additional experiments conducted. There are some remaining issues that need to be addressed, as outlined below:– The additional explanation of the analysis now provided by Figure 2 is welcome, but the layout and labelling of the main result in Figure 3 is still rather taxing on the reader and suboptimal for conveying the main result ('shape invariance'). In fact, Figure 3 is now less focussed as a result of including more data. Readers currently have to memorize which species and expression each of the 4 'style parameters' ("P1(e,s) – P4(e,s)") correspond to, and then parse a grid of 40 such plots in order to discern which they should be visually comparing. The previous iteration of this figure (previously Figure 2) with one column for macaque avatar and another for human avatar made this a bit easier. Minimally, the authors could improve the labelling of plots in Figure 3 to help readers parse the large array of heatmaps.

The reviewers’ criticism seems justified and is obviously a consequence of trying to squeeze so much more data in a single figure. We have now tried to split the figure into multiple ones, and we have tried to improve the labelling of the individual panels to make the figure more easily understandable. Specifically:

– We have split the conditions without and with equilibration and the data from the occlusion condition in three separate figures. Each of them has, in addition, the corresponding color plot of the significance levels belonging to the tests for differences between the multinomial classification distributions between the human and the monkey avatar.

– We have added an additional list that specifies clearly the association of the different discriminant functions π with the class labels (‘Human Angry/Threat’, ‘Human Fear’, ‘Monkey Angry/Threat’, ‘Monkey Fear’).

– We have adjusted the numbering of the figures in the text and updated the captions.

– The authors continue to insist that their macaque avatar achieves "the best known degree of realism" to date. There does not seem to be empirical evidence to support such a claim, since the authors' previous work (Siebert et al., 2020) did not compare their own avatar to other groups' avatars. At least one other group has found similar looking preference results for their own macaque avatar (Wilson et al., 2020).

Following the reviewers’ advice, we have carefully reworded the relevant section in the text to avoid the mentioned overclaims. The avatar by Wilson et al. is not dynamic and does not show an uncanny valley effect, but it results in similar looking times as original videos. We have included this reference and refined our wording. The modified text reads now:

‘The used dynamic head models achieve state-of-the-art degree of realism for the human head, and to our knowledge we present the only highly realistic monkey avatar used in physiology so far that is animated with motion capture data from real animals. […] In another study monkey facial avatar motion was controlled by parameters derived from video frames^35^. A further study showed similar looking times for static pictures of monkeys and avatar faces (without facial motion)^40^.’

In our view, this formulation is not an overclaim and should be justified by the data that we have provided. If reviewers should keep having problems with this paragraph, we will probably take it out completely, since it is not central for the paper.

– The links provided to online data and stimuli (https://hih-git.neurologie.uni-tuebingen.de/ntaubert/FacialExpressions) appear not to be publicly accessible at present. Will this be changed before publication or does access need to be requested? If the latter, instructions for obtaining access should be provided.

We have updated the links and hope that the material now is accessible. Please let us know if there remain any problems.